# Barriers to Exclusive Breastfeeding for Mothers in Tswelopele Municipality, Free State Province, South Africa: A Qualitative Study

**DOI:** 10.3390/children10081380

**Published:** 2023-08-13

**Authors:** Simthandile Rebecca Quebu, Daphne Murray, Uchenna Benedine Okafor

**Affiliations:** 1Department of Public Health, University of Fort Hare, 5 Oxford Street, East London 5201, South Africa; 202015436@ufh.ac.za; 2Department of Nursing Science, University of Fort Hare, 50 Church Street, East London 5201, South Africa; dmurray@ufh.ac.za; 3Faculty of Health Sciences, University of Fort Hare, 5 Oxford Street, East London 5201, South Africa

**Keywords:** exclusive breastfeeding, barriers, mothers, infants

## Abstract

Despite the numerous advantages of exclusive breastfeeding (EBF), the practice remains infrequently adopted in certain countries and is also associated with context-specific obstacles. Consequently, this study explores the experiences and opinions of mothers about the barriers and support systems of exclusive breastfeeding (EBF), in a bid to promote this practice in the Tswelopele Municipality of the Free State Province of South Africa. In-depth individual, semi-structured interviews were conducted with 16 mothers, using an audio recorder after receiving their permission to record the interviews. The analysis of the collected data revealed that opinions clustered around four topics: mother-related barriers to EBF, baby-related barriers to EBF, support systems to enhance EBF, and complications caused by barriers to EBF. The findings from these themes and sub-themes imply that the maternal factor is strongly influenced by other factors regarding the success or failure of this practice. These include extreme breast discomfort, maternal illness, the fear that the mother’s milk supply is insufficient, a lack of understanding about EBF, and the influence of different cultural factors. When mothers encounter difficulties with breastfeeding, these challenges may impede their ability to practice EBF. In addition, while some participants were eager to breastfeed their babies and continued EBF for a period of six months, their infant’s health and behavioral issues prevented them from doing so. Some of these problems included infant sickness and crying. From the prenatal to the postnatal period, mothers and their families should have access to breastfeeding education and counseling, along with sufficient time to make informed infant nutrition decisions. During counseling sessions, conversations with these stakeholders should focus on fostering a realistic understanding regarding what to expect when breastfeeding for the very first time, debunking breastfeeding-related misconceptions, and addressing inaccurate information and concerns. In addition, health professionals must be empathic and respectful of the mother’s traditions and cultures and must also educate mothers and their families on the importance of EBF. Our list of themes and sub-themes could be utilized to enlighten exclusive breastfeeding challenges and potential mitigation efforts, not only in Tswelopele Municipality, South Africa but also in a number of other geographical contexts.

## 1. Introduction

Exclusive breastfeeding (EBF) is known to effectively reduce infant and child morbidity and mortality rates [1,2,3]. This has numerous advantages for both the mother and the baby. Additionally, EBF reduces the risk of ovarian cancer and prevents unintended pregnancies [4,5]. Moreover, EBF has a long-term benefit in terms of BMI, as it prevents the accumulation of excess fat and, consequently, obesity [6]. Considering these advantages, initiatives to strengthen the practice of EBF are essential for promoting maternal health outcomes that are favorable.

Regardless of evidence supporting the clinical and medical importance of EBF, its implementation poses challenges in developed as well as developing nations [7,8]. While EBF is globally low at 35%, the percentage is even lower in Sub-Saharan Africa, ranging from 22% to 33% [9]. Considering EBF’s numerous advantages, this number is shockingly low. Despite the South African government’s efforts and attempts to promote EBF, the country’s EBF statistics remain significantly low (32%) [10]. Compared to the national target of 9%, the proportion of children younger than five who have died from severe acute malnutrition (SAM) was particularly high in two specific provinces, with a marginal increase in the Free State (18.3%), Lejweleputswa (15.8%), and the Tswelopele sub-district (16.7%) between 2012/13 and 2016/17 [11].

Barriers associated with EBF are multifactorial and have been extensively discussed in the literature. They include unemployment [12], caesarean deliveries [8], a lack of awareness about EBF [9,13,14], and contradictory family advice on EBF [14,15,16]. Other impediments to exclusive breastfeeding include the mother’s desire to return to full-time employment after maternity leave [3,15,17,18,19], and social and cultural factors, beliefs, and practices [3,9,14,16,20,21]. Promoting maternal health initiatives within the healthcare delivery agenda is essential for encouraging exclusive breastfeeding. To be effective, however, these approaches require an array of community, government, and institutional efforts.

Furthermore, the World Health Organization [22] endorses the promotion and safeguarding of EBF for the first six months of an infant’s life; furthermore, it recommends the addition of complementary foods to the diet at six months, along with breastfeeding, as this method has proven to be the most effective feeding practice to prevent infant-related illnesses and mortality worldwide, regardless of the HIV status and socioeconomic status of the mother. Notably, South Africa has instituted many policies, initiatives, strategies, and recommendations to promote, protect, and support breastfeeding [11]. These include the strong commitment of the National Department of Health (NDoH) to advance both the infant and young child feeding policy (IYCF) and the cause of maternal nutrition in the country, which led to the 2011 Tshwane Declaration of Support for Breastfeeding in South Africa. The government also revised the IYCF policy to motivate HIV-positive mothers to breastfeed their infants exclusively for the first six months, while receiving ART to prevent mother-to-child transmission (MTCT) and to continue breastfeeding until their babies are one year old [23]. The NDoH Roadmap for Nutrition in South Africa 2013–2017 [24] and the National Strategic Plan for Maternal, Newborn, Child, and Adolescent Health [25] are other initiatives. In addition, nurses were urged to educate first-time mothers about breastfeeding throughout antenatal care and during labor [23]. Other breastfeeding initiatives include Kangaroo Mother Care (KMC), MomConnect for pregnant women and mothers [26], the Mother-Baby-Friendly Hospital Initiative (MBFHI) [27] launched in 1991, and the KwaZulu-Natal initiative for Breastfeeding Support (KIBS) [28]. Notwithstanding these laudable drives, they demonstrably fail to account for the influence of family, community, and the workplace [12]; the 2016 South Africa Demographic and Health Survey reported that only 32% of infants younger than six months received EBF. Moreover, South Africa reportedly has the lowest breastfeeding rate in Africa [29]. Even before the increase in HIV prevalence, EBF was rare in South Africa [30]. Sociocultural, economic, and healthcare system-related concerns are associated with insufficient lactation practices in South Africa. Social conditions, such as a lack of income, reliance on food purchases, young mothers’ emotions about breastfeeding, and cultural beliefs were the primary determinants of mothers’ breastfeeding behaviors [30,31,32]. The breastfeeding process also excludes fathers. Other studies have alluded to the lack of support and involvement of the father figure in the breastfeeding agenda as an impediment to the adoption of EBF, along with other poor breastfeeding markers in South Africa [30]. Additionally, conflicting breastfeeding messages from family members/friends, healthcare workers, low socioeconomic indicators, and the fear of transmitting HIV to the newborn infant hinder efforts to exclusively breastfeed in South Africa [33,34,35]. In this regard, it is crucial that healthcare professionals provide mothers with pertinent EBF education and counseling [9,13,36,37]. Understanding the context-specific barriers to the practice of EBF is essential in order to develop the most effective support approaches to ensure its increased adoption. To the authors’ knowledge, only one published study on EBF has been carried out in the province of the Free State [12]. In light of this gap in the literature, this study explores the experiences and opinions of mothers about the barriers to and support systems of exclusive breastfeeding to increase the practicing of EBF in this region.

## 2. Study Setting

This research was conducted in Tswelopele Municipality, Lejweleputswa district, Free State Province, South Africa. This rural municipality has an estimated population of 47,625 people, with significant poverty and low education levels. Consequently, the vast majority of its inhabitants depend on agriculture for survival, while government grants are a main source of household income [38].

## 3. Research Design

A qualitative, explorative, and descriptive research design was employed to explore the perspectives of mothers concerning barriers to EBF practices, in addition to the support systems available to strengthen exclusive breastfeeding in the context of Tswelopele Municipality.

## 4. Participants and Sampling

The participants were 18- to 40-year-old mothers, with infants aged 6 to 12 months, who accessed infant and child healthcare services at three conveniently selected public healthcare facilities in Tswelopele Municipality. The participants were mothers who had visited health facilities for routine infant healthcare screening and maternal counseling. Conversely, purposive sampling was applied to select these mothers because they were likely to have recently completed the period in which EBF may have been practiced, making it simple for them to recall their specific breastfeeding practices.

## 5. Data Collection

Sixteen participants were selected for individual semi-structured interviews. After obtaining each participant’s consent, the primary investigator used a tape recorder to record the sessions. All the women volunteered and signed the informed consent form, willingly. The in-depth face-to-face individual interviews took place in specified private rooms within each health facility. The interviews lasted between 45 and 50 min, and data was collected in Sesotho before being translated and transcribed into English. The compilation of data spanned one month in October 2021 (twenty working days). Once data saturation was attained, the researcher terminated the interviews.

The study utilized questions that the researcher had previously formulated, based on the literature. On the basis of each participant’s responses, further questions were posed. The following questions were asked (Table 1).

## 6. Ethics

The University of Fort Hare’s Health Research Ethics Committee granted ethical approval for this study (Ref: 2021=06=12 QuebuS). In addition, the Research Ethics Committee of the Free State Department of Health, the Lejweleputswa District Department of Health, and the clinic administrators of the respective clinics permitted the study, as did the CEO of the hospital. It was made clear to participants that participation was voluntary and that there were no penalties for those who wished to withdraw at any point.

## 7. Observing Trustworthiness

The credibility of the data was established by creating a relationship of trust with the participants by outlining the research objectives and methodology and by interviewing them for an extended period of time. Multiple techniques were utilized to collect data, including the use of an audio recorder to capture data directly from the participants during interviews. In addition, each stage of the data collection process was meticulously documented so that each interview could be conducted using the same procedure. All sixteen participants utilized the same semi-structured interview guide, and the audio recordings were confirmed. In addition, an audit trail was established in order to determine the verbatim narratives, categories, and subcategories of the themes that emerged from the data analysis.

## 8. Data Analysis

The data were coded using a general inductive technique and repeated until no new concepts emerged. The process of data analysis was guided by Tesch’s eight-step approach [39]. The analysis of the data began with the transcription of the interviews. The initial data set was transcribed by repeatedly perusing it and listening to the audio recordings of the interviews. Then, common themes were identified, and the newly collected data were compared to the previously collected data to determine which themes were confirmed or not supported by the emergent information. The identified themes, as well as the sub-themes, were then categorized, coded, and forwarded to an independent coder for review.

## 9. Demographic Profile of the Participants

All the participating mothers were Black, and their ages ranged from 18 to 44 years, with the majority aged between 18 and 29 years. All the participants were unemployed and not in school, with most having dropped out before completing their secondary education. Many babies were born at the health center; however, not all were initiated into breastfeeding within the first hour. Notably, the socio-demographic characteristics of the respondents in our study are comparable to those of the population. The majority of rural women in the study context are poor and have a low level of educational attainment. In addition, their primary occupation is subsistence farming, and the majority of them rely on a government grant or stipend of R500 (approximately USD 29) per month as their household income. Probably because the majority of them are young and likely dropped out of school due to pregnancy, they are either unmarried or are residing with their partners. According to Hall et al. [40], South Africa has a significant percentage of single mothers, with just over 60% of children born in 2017 having no registered father. Table 2 provides a breakdown of the participants’ demographic information.

The data analysis revealed four major themes: mother-related barriers to EBF, baby-related barriers to EBF, support systems to enhance EBF, and complications caused by barriers to EBF (Table 3).

## 10. Theme 1: Mother-Related Barriers to EBF

Some mothers reported that they were able to effectively initiate breastfeeding while in the hospital. However, once they returned home, they found it difficult to continue EBF for six months due to a wide range of factors, including their opinions regarding having a low breast milk volume, the baby crying, and the baby being restless. Some mothers indicated that they lost interest in breastfeeding and decided to introduce more substantial foods earlier than recommended. Maternal illness was another reason why some participants never received adequate EBF education.

### 10.1. Sub-Theme 1.1: Extreme Breast Pain When Breastfeeding

Some of the mothers who participated in this study reported breast issues, such as breast pain, while lactating. The participants’ responses regarding breast issues that prevented EBF were as follows:

“My breastfeeding experience was painful. Really, when you start breastfeeding, your breast is so painful… From the nipples especially, and even as the baby was suckling and pulling the milk from my breast, it was so painful”. (Participant 5)

“Iyoh! For me, it was really painful, especially when he would grab the breast and start sucking it…” (Participant 10)

“I felt so much pain that I wished I did not need to breastfeed… I really just wanted to quit”. (Participant 10)

Some participating mothers reported having difficulty while breastfeeding due to sore breasts and sore nipples while their infants suckled. These participants quickly lost interest in breastfeeding and felt pressured to introduce solid foods and other fluids prematurely. However, discomfort was not their only reason for discontinuing breastfeeding, as a few also reported that certain breastfeeding intervals left them physically exhausted.

“I am a person who likes to sleep, so when he drinks at night, he would wake me up, and I would be tired”.(Participant 6)

In terms of the baby’s feeding intervals, this research revealed that (extreme) breast pain serves as a barrier to EBF, as do feelings of fatigue and weakness.

### 10.2. Sub-Theme 1.2: Mother’s Desire to Introduce Solids Prematurely

Most of the participants introduced other liquids and solid foods too early because they believed that breast milk solely could not provide adequate nutrition for six months. Two participants made the following assertions:

“Because he was not getting satisfied and full with just breast milk”.(Participant 8)

“Because I thought she was not getting satisfied and full with just breast milk”.(Participant 9)

Others indicated that they perceived certain behaviors as a sign that their breast milk was insufficient for satisfying their infants’ hunger, prompting them to introduce other liquids and solid foods too early. The following quotations provide support:

“Because she drank so much on me, her appetite was big, and I did not think my breast was producing enough milk”.(Participant 7)

“Every time I held him, he would start opening his mouth, he was crying, and drinking breast milk quickly after another feed, so my mom also agreed he was not getting full, and she said I must buy Purity and add that”.(Participant 10)

Additionally, the family members of the mothers encouraged them to introduce solid foods early.

“… His father also suggested that the baby was too old to be just drinking on my breast and water, we should add baby foods in order for him to be satisfied”.(Participant 5)

“… Both the baby’s father and myself did not see anything problematic with starting solid foods; as a matter of fact, the father was also very keen for baby to take other foods”.(Participant 1)

“…my mom was like, no way this child doesn’t get full with the breast as he should, so she went to buy a sachet of Cerelac and prepare it with some water, she prepared it in a small cup, and then we fed the baby two teaspoons of Cerelac, he ate very well as though he was an old baby and fell asleep after that, then he was fine”. (Participant 2)

### 10.3. Sub-Theme 1.3: Mother’s Diseases

Another significant maternal concern that hindered exclusive breastfeeding was the mother’s sickness. Two participants provided an explanation, as follows:

“I fell sick, and I was told to stop breastfeeding”.(Participant 10)

“So they told me they going to put an injection on me that is going to make me calm down, but it’s harmful for the baby, and it can be transferred to her through breast milk, so I had to stop breastfeeding at that time, and they gave her Prenan (formula milk) …”(Participant 14)

### 10.4. Sub-Theme 1.4: Ignorance of the Practice of EBF

Some mothers in this study were unaware of the practice of exclusive breastfeeding and its benefits. Some participants neglected to perform EBF because they lacked EBF-related knowledge; consequently, their lack of knowledge influenced their behaviors in this context. Furthermore, a number of participating mothers were unable to breastfeed for six months, despite receiving education on the subject at their respective healthcare facilities. Therefore, ignorance of EBF and its advantages is an impediment to the effective utilization of the practice; nevertheless, understanding EBF does not inherently translate to mothers practicing it. The following are examples of mothers’ comments about their understanding of EBF:

“No, I did not ever know about that practice”.(Participant 1)

“No, I have not heard. What I have heard of was giving a bottle as in formula milk, but nothing was mentioned with water”.(Participant 7)

“No, I did not ever know about exclusive breastfeeding”.(Participant 10)

“No, I did not. But my grandmother told me that for a baby to grow well, they have to breastfeed at least a year, but about exclusive breastfeeding, I did not really know”.(Participant 14)

In addition, participants stated that timely education would have enabled them to implement EBF.

“I think, had I known in time that I should give breast milk only and no other drinks and no other foods for the first six months of life, I would have avoided even thinking of trying to give formula milk, even to start giving water”.(Participant 7)

Some, however, did not continue EBF for six months, despite receiving sufficient information during their prenatal visits and labor.

“I heard at the hospital when I was going to deliver, they said I must give breast milk only for six months and to never give other foods or drinks, including water and formula milk, before then”.(Participant 9)

“… I figured out my breast did not have enough milk, and he doesn’t get full. So I decided by myself to make the mashed potatoes in addition to breast milk, and when I did give him the mashed potatoes, he would calm down and be normal”.(Participant 9)

### 10.5. Sub-Theme 1.5: Traditional Beliefs such as “Mohlala”

In addition, participants indicated that sociocultural beliefs and influences impede the practice of EBF. Mothers are persuaded to believe that breast milk is connected to cultural beliefs or practices, which is known as “mohlala”. When an infant is crying excessively while being breastfed, it is a sign of ill omen, and the mother should immediately discontinue breastfeeding. When asked about the causes of premature breastfeeding stoppage, one respondent stated:

“When I would give him my breast, he would cry a lot, and even at night, he started crying a lot. So, during my early lactation days, I was cared for by my grandmother, and upon hearing this unusual cry of the baby when I was trying to breastfeed the baby, she told me my breast has ‘mohlala’ and I must stop breastfeeding the baby, you know how these ‘grannies’ are. It’s these traditional things, like a bad omen, she did not explain what it really means, you know how elderly people can be, she just said it can’t happen that every time I put the baby on breast, he cries so much, it must be that, and I must stop the breastfeeding”.(Participant 5)

She was persuaded that “mohlala” has everything to do with the cultural convictions of her family’s elders, despite not being informed of its meaning.

“... She did not explain what it really means; you know how elderly people can be… It’s these traditional things, like a bad omen”.(Participant 5)

## 11. Theme 2: Baby-Associated Barriers to EBF

Apart from the aforementioned obstacles associated with mothers, infant-related factors also influence the success or otherwise of breastfeeding. Although some participants were eager to breastfeed their babies while continuing EBF for six months, their infants’ medical and other behavioral conditions prevented them from doing so. These problems included the infant’s refusal to breastfeed, sickness, vomiting, crying, and restlessness. The following two subthemes supported Theme 2 and will be discussed in further detail below.

### 11.1. Sub-Theme 2.1: Baby’s Refusal to Breastfeed

Several mothers reported that although their infants had initially started breastfeeding successfully, they eventually refused to suckle on their breasts; as a result, the mothers felt an urgent need to find alternative feeds.

“Next day at home, when I was breastfeeding him, he was refusing the breast. I tried giving him the bottle with just water as well, but he was refusing the dummy as well”.(Participant 2)

“He was just drinking on my breast and no other things for two weeks, but then he started not wanting my breast milk anymore, so I bought him formula milk”.(Participant 5)

In instances where infants refused to breastfeed, participants indicated that their initial maternal inclination was to seek an alternative nutrition method, thereby limiting their ability to breastfeed exclusively. In addition, a few mothers reported that their infants were unable to grasp the nipple and latch on, which may have contributed to their refusal of breast milk.


*“When I put the breast to him, he would refuse it by not suckling”.*
(Participant 2)


*“Randomly, when I was breastfeeding, he started crying, spitting the nipple out, and pushing my breast away, and would cry a lot.*
*”*
(Participant 13)


*“ Never latched on me, only cup-feeding throughout that week to when he got discharged … I wished he was feeding directly on my breast. Even when I got home, it’s only then my sisters were trying to teach me to breastfeed.*
*”*
(Participant 16)

The participating mothers also stated that such challenges may have caused their infants to become restless and refuse to breastfeed, which could pose a hindrance to EBF.

The majority of mothers also stated that their infants’ inexplicable crying concerned them so much that they felt compelled to discontinue EBF prior to six months of age. The following responses were recorded:


*“But the problem is when the baby starts crying now, it’s hard for you as a mother; it’s hard for you just look at your baby without wanting to prepare something for your baby”.*
(Participant 2)


*“It was the baby crying a lot that also made me think he was not being satisfied, so I also just introduced Nestum and baby foods with formula”.*
(Participant 5)

### 11.2. Sub-Theme 2.2: Sick Babies

In addition to the baby ailing, the mothers in this study also mentioned illness as a barrier to exclusive breastfeeding for six months. Participants expressed their experiences in the following manner:


*“I was troubled with how the baby was; I actually thought maybe he was sick, yet I didn’t understand because the doctors discharged us from the hospital with him very well. But now, when we get home, the issue is the baby is now starting to cry, and he doesn’t look fine”.*
(Participant 2)


*“At the hospital to breastfeed? She took three days before she was breastfeeding…. Yes, my breasts had milk, but the nurses made a mistake; they thought I was HIV-positive and I was negative, and I do not know how they made that mistake. Because when they discharged me, the baby was still admitted”.*
(Participant 15)


*“He did not breastfeed at all after birth; after the baby was born, they just took the baby away from me without any explanation. For the whole day, we were separated without explanation; meanwhile, the baby was in ICU. I only learned the next day. When I went to see him, the nurses in ICU asked me to try to express breast milk, but I was struggling at first. Eventually, it came…they were preparing her formula milk”.*
(Participant 16)

Furthermore, a few mothers indicated that their ill infants had a low level of interest in breast milk. One participant stated:


*“She vomited at one feed, and the second one, also immediately, she vomited. So my grandmother and her paternal grandmother told me, in that case, the breast milk was not good for the baby; it’s going to make him sick, even in future, I must never breastfeed”.*
(Participant 16)

## 12. Theme 3: A Support System to Enhance EBF

During their interviews, participants made a number of suggestions to increase the uptake of EBF for six months. Among these was the need to provide emotional support to breastfeeding mothers, in order to ameliorate maternal stress, as well as an increase in exclusive breastfeeding health education in primary healthcare facilities and other health institutions. The following subsections detail the supporting subthemes.

### 12.1. Sub-Theme 3.1: Health Education: Spelling Out the Benefits of Breastfeeding

As previously stated, the study participants suggested that planned health education by nurses might help mothers in implementing EBF for the first six months of their infants’ lives. In particular, healthcare professionals should educate mothers and emphasize this practice during prenatal clinic visits. The following are examples of responses from the participants:


*“I think if the nurses at the clinic taught us as soon as the baby is born or even better during pregnancy about not giving water or baby foods, even formula milk, before the baby is six months and to give breast milk only during that period would really be of help and supportive, instead of the way I attained this knowledge later, when the baby was born and already drinking water. At that time I was even trying to give her formula milk, only she refused the bottle”.*
(Participant 7)


*“I think by communicating and educating lactating mothers clearly because some really don’t know”.*
(Participant 6)


*“As we come to clinic, you can ask all breastfeeding mothers separately and address them specifically for like 5–10 min about EBF for six months and the importance thereof and ensure that they are fully informed”.*
(Participant 11)


*“Maybe publicly at the clinic, especially to pregnant women and mothers with young babies”.*
(Participant 15)


*“Maybe publicly at the clinic, sit them down all the mothers and explain the importance of EBF for six months. Call them aside and address them separately, especially the young mothers”.*
(Participant 16)

Other participants indicated that health education on topics such as how to express breast milk and the recommended maternal nutritional intake to boost breast milk production may motivate and help mothers regarding continuing EBF. Here are a few illustrations of such sentiments:


*“Maybe if I was told to express some of my breast milk in the bottle and then he would drink from it”.*
(Participant 13)


*“It would have been helpful if I was taught early enough to be aware of the kinds of food that help with milk production because I discovered late that when I drink tea and soft porridge, I produce more milk than when I eat other foods”.*
(Participant 8)

### 12.2. Sub-Theme 3.2: Reduction of Stress for the Mother

The participants cited tension as an additional cause of premature EBF cessation. In addition, they emphasized that having the support of family and friends or being a member of an organization who could support them can help to mitigate this issue. In addition, their responses suggest that carrying the burden of breastfeeding challenges alone makes mothers more likely to abandon EBF practice; therefore, it is crucial to reduce maternal stress in order to increase EBF practice. The following two responses confirmed this:

“I wish the father of my baby would have stood by my side and followed it up because he is the one who sent me to his family when the baby vomited my breast milk, and they immediately instructed me to stop breast milk”.(Participant 16)

“Seeing other people who can sustain EBF six months, then I can see that I also can… Close people around me would be more supportive”.(Participant 1)

### 12.3. Sub-Theme 3.3: Cost-Effectiveness: Mothers Saved from Buying Milk Formulas and Other Solids

Regarding the introduction of other foods or fluids, mothers in this study were concerned about the cost of infant nutrition supplies. Concerning their decision to discontinue breastfeeding, mothers cited the price of alternative foods to breast milk as their main concern. Certain participants expressed the following:

“They taught us that breastfeeding is best in the first six months. You just give breast milk, especially us unemployed mothers, because with me in my first pregnancy, the father left me, and there was no way I could even afford formula milk”.(Participant 11)

“...it was not easy for me to wean him because I wondered what else would my baby eat because I can’t even afford formula milk”.(Participant 12)

### 12.4. Sub-Theme 3.4: Encouraging Mothers to Be Patient and Persistent in Breastfeeding

The participants stated that, when it comes to exclusive breastfeeding, perseverance, as well as patience, can help surmount obstacles. In addition, they suggested that empowering breastfeeding mothers by teaching resilience techniques might motivate them to devote more time to EBF. In addition, interviewees stated that if a mother persists through challenges with breastfeeding for the sake of her infant, she might want to continue EBF for the recommended duration. Listed below are a few responses:

“If, maybe, I built up patience and endured longer, that could have helped me with EBF”.(Participant 10)

“… advise the mothers to be persistent and keep trying, even if they struggle at first with breastfeeding only, they must not just decide to stop, for the sake of their baby”.(Participant 4)

“Many times, I would build my courage and tell myself I should not give in to the temptation to start baby foods earlier than six months because they taught us at the clinic it’s good for the baby, and so I held it up”.(Participant 11)

## 13. Theme 4: Complications Created by Barriers to EBF

Various participants who had experienced breastfeeding difficulties that prevented them from sustaining EBF for six months expressed their concerns that failure to do so would result in additional complications, including infants not receiving the protective benefits of breast milk. Listed below are the supporting subthemes for Theme 4:

### 13.1. Sub-Theme 4.1: The Mother’s Failure to Bond with the Baby

Some participants were dissatisfied that, due to poor health, they were unable to breastfeed their infants from delivery, which prevented them from forming a bond with their children. A few mothers described the high quality of the time they spend breastfeeding their children. One of the participants articulated this experience as follows:

“I enjoyed how we would bond nicely”.(Participant 11)

Breastfeeding is a distinct bonding experience shared exclusively between a nursing mother and her infant(s). As previously stated, breastfed infants receive optimal nutrition and are nurtured by their mothers through close physical contact. When breastfeeding is effortless and uninterrupted, the well-being of both the mother and the infant is typically high [41]. However, the premature introduction of solid foods may reduce the amount of time a newborn spends nursing, thereby diminishing the benefits of this unique bond between mother and child.

### 13.2. Sub-Theme 4.2: Denying a Baby the Benefits of Breast Milk

Some mothers felt that they had no influence over EBF barriers. In addition, a few participants expressed regret and claimed that they would have made better decisions if the circumstances were different. The following are two of the recorded answers:

“It was a traditional thing for me; had it not been for my grandmother’s input and advice, I would have continued to breastfeed up to 18 months at least and just give breast milk for six months”.(Participant 5)

“I looked forward to seeing my baby grow and develop healthily because breastfeeding babies grow healthy and their weight is always good”.(Participant 14)

In support of the aforementioned comments, a number of participants acknowledged that their newborns were susceptible to a variety of health issues that are not present in infants who are exclusively breastfed. They indicated that they were aware of the implications of introducing complementary foods prematurely. The following are two of the health concerns that they mentioned:

“…but also, I was aware that for his age, I must start giving water so that he doesn’t get constipation; also, the sisters had told me that I shouldn’t give water before the end of six months of-age of the baby, but because I had given my baby other foods, then I decided I’m going to give him some water”.(Participant 2)

“What I loved most was that he was developing well and weight gaining was also good, but when he stopped breastfeeding, he got sick for two weeks, when I stopped breastfeeding”.(Participant 8)

## 14. Discussion

This qualitative study investigated the barriers to and support systems for exclusive breastfeeding. Four major themes emerged from our interviews: mother-related barriers, baby-related barriers, EBF support systems, and complications resulting from EBF. Under each theme, we highlighted the barriers and strategies that could hinder or assist future EBF practice. In the short term, these results will aid efforts to address infant feeding practices in Tswelopele Municipality, South Africa.

The findings of this study indicate that mothers have difficulty with EBF and that maternal factors are among the primary determinants of whether EBF would be successful. This idea is consistent with the findings of Jama et al. [42], who found that despite successful breastfeeding initiation at hospitals after birth, some mothers find breastfeeding challenging. This resonates with the findings of this study; some mothers did not maintain breastfeeding for six months despite receiving sufficient information from nurses during prenatal visits and labor at health clinics. However, understanding EBF does not automatically result in mothers practicing EBF. As indicated earlier by one of the participants: 


*I heard at the hospital when I was going to deliver, they said I must give breast milk only for six months and to never give other foods or drinks including water and formula milk before then… I figured out my breast did not have enough milk, and he doesn’t get full. So I decided by myself to make the mashed potatoes in addition to breast milk, and when I did give him the mashed potatoes, he would calm down and be normal”.*
(Participant 9)

In addition, they believed that EBF did not effectively satisfy their infants’ hunger at home; consequently, they were less likely to practice EBF. Moreover, these participants elaborated on the difficulty of sustaining EBF for six months and provided a variety of explanations.

Furthermore, a few participants reported that breastfeeding was challenging because of painful breasts and nipples; consequently, they lost interest in the practice and also felt compelled to introduce solid foods and other fluids to their infants prematurely. However, in addition to breast pain, fatigue and exhaustion also contributed to their lack of motivation. Childcare, domestic chores, and the care of other children may have contributed to the mother’s boredom; however, these issues were not investigated. Nevertheless, anecdotal evidence suggests that mothers are rarely assisted with lactation, particularly in terms of receiving assistance with other family responsibilities. At most, they received breastfeeding advice from close family members and relatives, which can be misleading and even incompatible with the principles of good lactation practices. Several of the mothers stated that certain breastfeeding intervals would physically exhaust them. According to a study by Gianni et al. [43], about 8–10% of breastfeeding mothers begin mixed feeding due to the excruciating pain felt during breastfeeding. This discomfort is commonly brought on by improper latching on and positioning of the newborn during feeding, but breastfeeding may additionally result in nipple cracks, mastitis, and engorgement. As previously mentioned, research indicates that maternal factors such as drowsiness and fatigue can impede exclusive breastfeeding in a disproportionately high number of cases [43].

In addition, most of the mothers in this current study indicated that they introduced supplementary foods too early because they thought breast milk alone would not provide sufficient nutrition for their infants’ first six months. In addition, their infants’ behaviors, such as unexplained weeping and increased appetite, were interpreted as signs that breast milk alone was insufficient to satisfy their hunger. Consequently, they introduced other liquids and solid foods prematurely. Previous studies have shown that the most commonly reported barrier to EBF is an alleged lack of sufficient breast milk to nourish the infant [42,44]. The justifications for this belief include the baby’s incessant crying and/or desire to breastfeed for an extended period of time. On the other hand, a few mothers reported being persuaded by family members to start solid food too early, a finding consistent with an Ethiopian study by Deme et al. [45], which found that 32.9% of babies were introduced to other foods and fluids by the age of three months, primarily by mothers who gave birth at home and those whose family members influenced them to introduce mixed feeding. Participants also reported that maternal illnesses compelled them to discontinue breastfeeding in order to secure the well-being and health of their infants. This supports the findings of Thepha et al. [14] that physical complications for the mother, such as breast problems, postpartum depression, cesarean section birth, smoking, and drug addiction, serve as barriers to exclusive breastfeeding.

Notably, the potential causes of insufficient breast milk cited by the mothers in this study may be attributable to poverty and improper nutrition. Nevertheless, according to the World Health Organization, feeds of breast milk could be enhanced by feeding the infant every two to three hours [46], which may be sufficient for the mother and the infant. Similarly, the barriers cited by the mothers in this study, and their belief that the crying of a newborn in the early stages of birth indicate hunger, may mislead the mother into believing that she produces insufficient breast milk, resulting in the tendency to introduce other foods or infant formula. It is, therefore, essential to encourage mothers to develop confidence in their capacity to generate enough milk for their infants [47]. Thus, antenatal settings are ideally suited for such breastfeeding-related talks and discussions, to help mothers and healthcare professionals avoid misinterpreting early infant behaviors (crying, short nighttime periods of sleep, and reluctance to be breastfed) as pathological during breastfeeding [48,49]. Several women who participated in the study indicated that they were unaware of the practice of EBF and its benefits and that if they had been taught and provided with EBF information earlier, they would have been empowered to implement and continue the practice. Studies have shown that mothers who are knowledgeable about the benefits of EBF have a positive attitude toward sustaining exclusive breastfeeding using various methods, such as breast pumps and expressed breast milk [50,51]. As part of South Africa’s efforts to encourage and promote exclusive breastfeeding, nurses are required to educate first-time mothers about lactation in antenatal care and during labor [23]. Despite their health workers’ recommendations to exclusively provide breast milk for six months, mothers feel compelled to introduce solids or other fluids [52]. However, this present study also revealed that even when health workers educate mothers appropriately, the latter do not always apply this information. This is comparable to the findings of Mgongo et al. [21] in Tanzania, who reported that the majority of breastfeeding mothers who were not informed by their healthcare providers that breast milk is optimal for the baby’s health in the first six months, and who were not taught that breast milk is the only food a child should consume in the first six months, were unsuccessful in the practice of EBF. This, however, contradicts the findings of other studies in Kenya, which discovered that even though mothers were taught and have knowledge about EBF, it is not practiced [53,54]. Additionally, a recent systematic and meta-analysis review study revealed that, despite these women’s good level of knowledge and positive attitude towards EBF, there is a substantial gap regarding EBF practice; thus, maternal and child health services must be strengthened [55].

One participant added that her family’s traditional beliefs prevented her from practicing EBF. This suggests that sociocultural attitudes are crucial in determining mothers’ infant nutrition patterns and could serve as a barrier to exclusive breastfeeding. A previous study found that although the majority of breastfeeding mothers were opposed to the use of traditional medicine or practices, their family elders proposed their use for the protection of their infants; participants from rural regions regularly reported using traditional medicine or practices [44]. Numerous studies have reported that women engage in complementary infant feeding practices because of cultural beliefs and family pressures [3,13,16,20,21,31,33,38]. The majority of African societies are unfamiliar with EBF because it contradicts their deeply held cultural beliefs and customs [56]. Given that women are closely tied to their menfolk and other family members and acquaintances, EBF decisions should be made with their input [56]. In addition, there is typically no formal lactation environment; thus, mothers are free to breastfeed at any time and place [57]. Furthermore, in some societies, women who breastfeed publicly are not obligated to conceal their breasts [57]. Consequently, the lack of support from family and relatives may make it difficult for the mother to breastfeed exclusively. Family support is a key strategy for strengthening EBF; this support may be psychological, emotional, financial, or informational, and it can help the mother succeed in her quest for exclusive breastfeeding. There is a positive relationship between family support and exclusive breastfeeding, which increases a mother’s desire for and willingness to continue EBF [58].

While maternal factors also play a role in breastfeeding practices, infant-related factors further impact the efficacy of breastfeeding. This was demonstrated in the current study, in which some participants had been determined to breastfeed their newborns while continuing EBF for six months but stated that their infants’ health issues and behaviors prevented them from doing so. These problems may include the infant’s sickness and crying. Unexplained weeping in babies is frequently attributable to the aforementioned factors, resulting in difficulty in latching on and reluctance to be breastfed. Several mothers reported that, while their infants initially fed well, they eventually seemed reluctant to latch on. Consequently, the mothers felt an urgent need to provide alternative feeds. Many mothers reported that their infants were unable to grasp and suckle, resulting in a reluctance to breastfeed. Previous research has linked nipple difficulties to poor latching and placement [59]; however, a recent study [60] reported a contradictory finding. Instead of seeking alternatives to breastfeeding, mothers should be educated on how to overcome these physical obstacles to EBF via the correct practices. In addition, some parents reported that their infants’ uncontrollable sobbing, even after breastfeeding, caused them (and some family members) to provide their infants with additional fluids and solid foods. These findings support a Brazilian study by Amaral [61] that describes infants’ unexplained refusal of breast milk as an impediment to EBF, which necessitates that mothers introduce solid foods or other fluids, even if they had intended to continue with EBF for six months. In addition, some infants may have weak jaws, making it difficult for them to suckle, resulting in their persistent crying, causing their mothers to feed them various foods [14,57]. According to Sehgal [62], certain childhood illnesses may cause infants to refuse breastfeeding because of gastrointestinal symptoms. An earlier study by Murray and Christie [63] also indicates that approximately 50 percent of infants vomit at least once a day, from birth to three months of age; by four months, this proportion increases to nearly 70 percent. According to research, acid reflux is frequently associated with vomiting or increased physical activity [63].

The current study’s participants suggested that in order to promote EBF, clinics and hospitals should provide regular and comprehensive EBF-related health education and emotional support to breastfeeding mothers, in order to reduce maternal stress and promote the practice’s sustainability. In addition, they suggested that nurses should educate mothers, particularly during prenatal clinic visits, to encourage the women to breastfeed exclusively for at least six months. The healthcare system must provide regular infant nutrition guidance to mothers who are pregnant or breastfeeding [58]. This is in accordance with the recent UNICEF and WHO recommendations, which emphasize the importance of providing breastfeeding counseling to mothers, in order to enhance their breastfeeding practices [64]. According to Kavle et al. [49], antenatal and postnatal care providers are required to have the most up-to-date and essential skills in order to resolve EBF-related issues; hence, adequate training is necessary. In addition, the government must urgently develop and implement policies that protect and encourage EBF as a component of an intervention strategy [13]. Similarly, understanding the health advantages of breast milk can increase a mother’s desire to lactate and can prolong the duration of breastfeeding [14]. Some respondents maintained that being equipped with health education, including knowledge of how to express breast milk and improving maternal nutritional intake to increase breast milk production, can help increase EBF practice. Likewise, Motee and Jeewon [65] state that although expressing breast milk has its disadvantages, in comparison to the baby drinking directly from the breast, it is, however, still beneficial since mothers who are unable to breastfeed can express their milk. It is still important since it is the infant’s only opportunity to consume human milk. Additionally, the participants suggested that the mothers’ understanding and knowledge of the health advantages of breast milk for their infants could be of great assistance in implementing EBF. Desai et al. [66] found that the majority of mothers (84%) knew that EBF was essential, particularly during the first six months, because it reduces their infants’ risk of illness and provides them with sufficient energy for growth. This awareness of EBF-related benefits had a significant relationship with EBF use.

Furthermore, some participants stated that having the support of a loved one and belonging to a supportive group can help them overcome obstacles to EBF. The participants also found that carrying the burden of breastfeeding challenges alone led mothers to discontinue EBF; therefore, reducing maternal stress effectively can increase EBF practice. According to Thepha et al. [14], not only does family support have a positive effect on EBF but also supportive health care services, such as a six-month postnatal check, home visit follow-ups, nursing support, breastfeeding guidelines and education programs, breastfeeding promotion programs, and reasonable access to healthcare during the prenatal and postnatal periods, which are all facilitators of EBF. Furthermore, Ganu [53] found a significant relationship between support networks, motivation, and breastfeeding. Lactating mothers with strong support networks are more likely to exclusively breastfeed their offspring.

This suggests that sustaining the financial burden of the early introduction of solid foods and other fluids can be challenging and nearly impossible, particularly for unemployed mothers. Since breast milk is free and is readily accessible compared to the costs of breast milk substitutes, its cost-effectiveness may motivate unemployed mothers to implement EBF. Mgongo et al. [21] indicate in their study that EBF saves mothers money because they do not have to purchase formula or cow’s milk for the first six months. In addition, because the child does not become ill as frequently as a result of EBF, the parents do not have to seek medical treatment as frequently, allowing the parents to save money.

Some mothers who participated in the current study believed they had no control over EBF-related factors. In addition, they acknowledged that as a consequence of the early introduction of complementary foods, their infants were more susceptible to a variety of health issues than EBF infants. According to Adda et al. [67], previous Ghanaian studies indicate a correlation between the early introduction of solid foods and liquids and the risk of increased morbidity and malnutrition among children. In addition, these authors state that EBF is associated with a reduced risk of childhood and adolescent obesity and cognitive impairment.

## 15. Study Limitations

We acknowledge this study’s limitations. Firstly, the purpose of the study was to explore and describe the perspectives of the study participants, for which we used a purposive sampling technique; consequently, the results may not be applicable to the entire population. Second, although we have identified a number of pertinent barriers to exclusive breastfeeding practices for infants under six months and also the necessary support systems to enhance this practice, we do not know the prevalence of these barriers. To ascertain this, additional research, along with alternative (quantitative) methods, are necessary. Thirdly, the birth order of the children and the gestational durations were not explored. Finally, we had originally planned to conduct focus-group discussions but decided against doing so due to the COVID-19-related risks associated with gathering groups together. It is possible, although unlikely, that additional themes emerged.

## 16. Implications of the Study

The findings of this study provide insight into the barriers to EBF in a rural, resource-limited setting and serve as motivation to develop context-specific strategies to address breastfeeding-related challenges and to improve the culture and practice of EBF in this geographic area. Empowering and supporting new mothers through explicit and specific health education regarding the practice, duration, and advantages of EBF may increase the uptake of EBF practice in women [68]. Additionally, mothers, along with their families, ought to have access to breastfeeding education and counseling from the prenatal to the postnatal stages. Thus, they will have sufficient time to make well-informed decisions regarding infant nutrition. Health professionals can also inform parents of the importance of exclusive breastfeeding, as well as the costs and dangers associated with not following the practice.

Moreover, during the counseling sessions, women and their families can have an open and frank discussion about what they can expect when they begin breastfeeding for the first time, as well as explore misconceptions, incorrect information, and concerns. Health professionals should also consider the ethnic and cultural backgrounds of mothers, be aware of the cultural customs associated with breastfeeding, and educate women and their families carefully about traditions that may impact breastfeeding. Furthermore, family involvement is essential because a supportive companion, relative, or friend is essential for breastfeeding success [69,70]. If a mother feels supported in her decision to breastfeed, she is more likely to feel confident and in control of her decision. Therefore, providing consistent and continuing health education on breastfeeding to family members (grandparents, elders, partners, and siblings) and friends will allow other family members who have breastfed in the past to discuss their personal experiences, while creating a breastfeeding-friendly environment for nursing mothers [36]. In this context, healthcare professionals are essential and can provide precise and customized health education regarding the practice of EBF, its duration, and its benefits. In addition, this study demonstrated that sociocultural beliefs play an important role in determining mothers’ feeding practices for babies and may serve as a barrier to EBF. Therefore, interventions that target sociocultural perspectives in order to clarify stereotypical misconceptions and falsehoods about EBF practices are necessary. Future research relating to this study should replicate comparable qualitative work in other contexts, both in Tswelopele Municipality, South Africa and in other countries, to establish differences as well as similarities in EBF practices. Furthermore, quantitative surveys to determine the prevalence of the identified barriers and mitigation measures are necessary.

## 17. Conclusions

In this study, the participants experienced multiple challenges that prevented them from breastfeeding exclusively. Specifically, two kinds of barriers were identified: mother-related and baby-related barriers. Nevertheless, since it is essential for mothers to bridge these gaps, breastfeeding health education for expectant women and their families, together with family support or support from close relatives for breastfeeding mothers, can offer an excellent support system for bolstering EBF practice. Lastly, this research demonstrates that these barriers result in difficulties such as a mother’s inability to form a bond with her infant and newborns not receiving the advantages of breast milk. From a broader perspective, we recommend the implementation of the context-specific interventions emphasized in this study to overcome the obstacles impeding EBF in this geographical setting.

## Figures and Tables

**Table 1 children-10-01380-t001:** Interview guide.

1. How would you describe your experiences as a mother regarding barriers to exclusive breastfeeding?
2. What have you heard about exclusive breastfeeding?
3. What was your experience of feeding a baby with breast milk only?
4. What was the best thing about exclusive breastfeeding?
5. What was the most challenging?
6. Why did you initiate complementary/other foods and drinks in addition to breast milk during the first six months postpartum?
7. What could have helped/encouraged you to feed your baby breast milk only for up to six months?
8. What support systems would you require as a mother to strengthen exclusive breastfeeding?

**Table 2 children-10-01380-t002:** Demographic profile of participants.

Participant	Age (Years)	Marital Status	Education Level	Age of Baby (Months)	Place of Delivery	Mode of Delivery	Breastfeeding Initiation Time Post Delivery
1	18	Single	Grade 11	12	Home	Natural	> 1 h
2	34	Single	Grade 9	6	Hospital	Natural	< 1 h
3	29	Single	Grade 10	10	Hospital	Natural	< 1 h
4	44	Married	Grade 5	8	Hospital	Natural	< 1 h
5	28	Single	Grade 9	9	Hospital	Natural	< 1 h
6	28	Single	Grade 9	8	Home	Natural	> 1 h
7	33	Married	Grade 11	9	Hospital	Natural	> 1 h
8	18	Single	Grade 8	7	Hospital	Natural	< 1 h
9	35	Married	Grade 5	12	Hospital	Natural	< 1 h
10	22	Single	Grade 10	8	Hospital	Natural	< 1 h
11	28	Single	College	6	Hospital	Natural	< 1 h
12	20	Single	Grade 4	9	Hospital	Natural	< 1 h
13	21	Single	Grade 8	6	Hospital	Natural	< 1 h
14	18	Single	Grade 9	11	Hospital	Natural	< 1 h
15	23	Single	Grade 9	9	Hospital	C/section	> 1 h
16	25	Single	Grade 12	9	Hospital	Natural	> 1 h

**Table 3 children-10-01380-t003:** Identified themes and sub-themes.

Themes	Sub-Themes
1. Mother-related barriers to EBF	1.1 Extreme breast pain when breastfeeding
	1.2 Mother’s desire to introduce solids prematurely
	1.3 Mother’s diseases
	1.4 Ignorance of the practice of EBF
	1.5 Traditional beliefs such as “mohlala”
2. Baby-related barriers to EBF	2.1 Baby’s refusal to breastfeed
	2.2 Sick babies
3. Support systems to enhance EBF	3.1 Health education: spelling out the benefits of breastfeeding
	3.2 Reduction of stress for the mother
	3.3 Cost effectiveness: mothers saved from buying milk formulas and other solids
	3.4 Encouraging mothers to be patient and persistent in breastfeeding
4. Complications caused by barriers to EBF	4.1 Mother’s failure to bond with the baby
	4.2 Denying the baby the benefits of breast milk

## Data Availability

Data available on request.

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
