# Peer review of "Barriers to Exclusive Breastfeeding for Mothers in Tswelopele Municipality, Free State Province, South Africa: A Qualitative Study"

_children, 2023, doi:10.3390/children10081380_

Round 1

Reviewer 1 Report

Abstract:

-        Regardless of the numerous advantages of exclusive breastfeeding (EBF), few mothers practice it (what it means for the authors  „few mothers“ – 44%)

-        this study examines exclusive breastfeeding (EBF) barriers and support systems (how the support system was examine ?)

-        Four major themes emerged (emerged or were selected by the authors ?)

-        that maternal factors are the primary contributors to the success or failure of this practice (interrogative construction - are maternal factors independent of the influence of other factors?)

-        while some participants were anxious to breastfeed the babies (how anxiety was assessed ?)

-        "refusal of infants to be breastfed and mother's milk is bad accept" is assessed by vomiting after feeding ???

Introduction:

-        from the introduction we learn very little about the specifics of exclusive breastfeeding in South Africa

-        According to Kavle et al. [24], antenatal and postnatal care providers are required to have the most up-to-date and essential skills in order to resolve issues that may arise; hence, adequate training is necessary. In addition, the government must urgently develop and implement policies that protect, encourage, and encourage the EBF as a component of an intervention strategy [14]. - the introduction should explain in more detail the social, cultural, economic, traditional, health and other specificities of breastfeeding in southern Africa, and the quoted sentences should be left for discussion

-        In light of this, this study investigates the barriers and support systems necessary to support and improve the practice of EBF in this region. - can it be said that the data analysis of the 16 respondents is research into the breastfeeding support system

Study setting:

-        from the population of from 47,625 people were selected 16 respondents - is this a representative sample

-        according to which criteria were chosen 4 public healthcare facilities to participate in the study

-        from which clinics mothers were chosen to participate in the study - if they were separated from clinics where mothers seek help, then the authors influenced the sample

Data collection:

-        according to which criteria the questions were formed

Results:

-        Table 2. Demographic profile of participants

a) whether the demographic profile of participants differs from the demographic profile of mothers who did not participate in the research

b) whether the factors highlighted in Table 2 influenced the responses of the respondents, i.e. whether differences in the responses of the respondents can be determined based on the differences in these factors

-     Table 3. Identified themes and sub-themes. - what were the criteria for selecting subtopics - for example, similar answers from more than 50% of respondents..

-     Theme 1: Mother-related barriers to EBF - whether "having a low breast

       milk volume" can be explained by "crying and restless infants" or is it a link presented by the respondents? (express more clearly)

-        Sub-theme 1.1: Extreme breast pain when breastfeeding - were the respondents familiar with the influence of proper breastfeeding technique on the pain during breastfeeding? did they use any procedures after breastfeeding to relieve nipple soreness?

-        Sub-theme 1.2: Mother's desire to introduce solids prematurely - do the authors have information about prenatal education of mothers? are there data in other works about the level of knowledge of mothers about breastfeeding ?

-        Sub-theme 1.3: Mother’s diseases  - pregnancy cannot be equated with illness;                   they told me they going to put an injection on me - who said he would give an injection that could harm the child

-        Sub-theme 1.4: Ignorance of the practice of EBF - can the authors describe the differences in the responses of respondents who had breastfeeding education and those who did not have such education

-        Sub-theme 1.5: Traditional beliefs like 'mohlala' - the concept of „mohlala“ and what the health service is doing about it should be explained in more detail

-        Sub-theme 2.1: Baby's refusal to breastfeed - the authors should definitely describe the "baby's refusal to breastfeed" syndrome in great detail and support it with data from professional literature

-        intolerance to breast milk, as evidenced by vomiting after breastfeeding. - authors should very carefully explain "intolerance to breast milk" which is recognized "by vomiting after breastfeeding", and they should support this with data from the literature

-        Theme 3: Support system to enhance EBF -  In the introduction, much more should have been said about the peculiarities of the health system and the way mothers are prepared for pregnancy and breastfeeding, as well as breastfeeding support programs organized in South Africa. In addition, the responses of mothers who have received breastfeeding education should be separated from those who have not. In this way, it is very difficult to evaluate the answers of the respondents.

-        Sub-theme 4.1: Baby’s failure to bond with its mother - rather one could talk about the mother's failure to achieve secure attachment rather than the child's failure. Please support any such claim, regardless of whether it reflects the author's position or the respondents' position, with references

Discussion:

-        some mothers experience breastfeeding challenging and humiliating - breastfeeding is a challenge for every parent, so not only the mother but also the father. We expect the father to support the mother, wanting the best for his child. The authors should explain in what way the test subjects experienced breastfeeding as humiliating.

-        few participants reported that nursing was challenging because of painful breasts and nipples - again, the authors do not say anything about the procedures for relieving pain in the nipples or how familiar the test subjects are with these procedures

-        authors should distinguish maternal exhaustion caused by breastfeeding from exhaustion due to a disturbed mother's sleep rhythm, exhaustion due to child care (except breastfeeding), exhaustion due to household duties, exhaustion due to care for other children.

-        if they are already talking about the exhaustion of the mother due to breastfeeding, the authors should also talk about the system of family support for mothers (help of partner, mother, sister, etc.). What are the customs and family roles in the country where the study was conducted?

-        mothers can get information about good or poor progress of their child from the pediatrician. If, nevertheless, they associate the infant's crying exclusively with unsuccessful breastfeeding, the authors should consider the option of other possible reasons, which are hidden behind the formal statement that the mother's milk was not sufficient for the child's development. If the system of family, health and social support for breastfeeding is weak, perhaps mothers are looking for an acceptable way to stop breastfeeding, and as a reason for stopping they offer a problem on the part of the child, in order to avoid condemnation from the environment.

-        it is not enough to say that mothers do not apply the acquired knowledge about exclusive breastfeeding. Possible reasons should be discussed. The reason is probably not the mother's desire to harm the child. Then why doesn't the mother provide the child with the best food. Other research clearly shows that the answer should be sought in the system of family, social, economic and health support. The authors limit themselves only to the immediate, although in conversations with women they were able to find out data for a deeper analysis.

-        include infants who refuse to breastfeed or who have an intolerance to breast milk - what is the probability that the authors of the 16 respondents encountered children who refuse to breastfeed or who have an intolerance to breast milk. Can the authors say according to the data in the literature that intolerance to breast milk is common?

-        Desai et al. [42] found that the majority of mothers (84%) knew that EBF was essential, particularly during the first six months, because it reduces infants' risk of illness and provides them with sufficient energy for growth. - the authors remain on the surface of the problem, they themselves cite examples of mothers who have been educated about breastfeeding and who stopped breastfeeding anyway. This clearly indicates that mothers' knowledge is not enough for successful breastfeeding. Breastfeeding is a challenge, it is demanding and the mother needs organized and systematic help to persist in breastfeeding, which in most children starts in the maternity ward. It is not enough to just tell mothers how important exclusive breastfeeding is, they need to be supported and helped in an organized way, through the family support system, through visits from the visiting nurse, through breastfeeding support groups, through social and economic support measures organized by state institutions.

Study limitations

       - we used a purposive sampling technique - it compromises the results

-        Finally, we had originally planned to conduct focus group discussions - this can explain the insufficient depth of the work, the insufficient processing of the respondents' statements, the consistent acceptance of their statements without requiring them to explain these statements, and even when they are contradictory or when the respondents themselves confirm that they acted contrary to the given instructions

Author Response

Comments and Suggestions for Authors

Abstract:

Comment:

Regardless of the numerous advantages of exclusive breastfeeding (EBF), few mothers practice it (what it means for the authors „ few mothers“ – 44%)

Response:

The sentence has been revised or rephrased thus: “Despite the numerous advantages of exclusive breastfeeding (EBF), it remains less widely practiced, and it is also associated with context-specific obstacles.”

Comment:

This study examines exclusive breastfeeding (EBF) barriers and support systems (how the support system was examine?)

Response:

As per qualitative study, it word ‘examines’ is replace with ‘explores’. As stated, individual, semi-structured interviews were conducted to explore the barriers and support systems to promote EBF practice.

Comment:

Four major themes emerged (emerged or were selected by the authors?)

Response:

The four themes 'emerged' from the analysis of the data. This term is commonly used in qualitative research reports.

Comment:

That maternal factors are the primary contributors to the success or failure of this practice (interrogative construction - are maternal factors independent of the influence of other factors?)

Response:

We stated that the results of our study demonstrate that maternal factors are the main contributing factors to the success or failure of exclusive breastfeeding, at least in the setting under consideration. Nonetheless, this question necessitates a quantitative investigation to determine statistically whether maternal factors are independent of other factors influencing EBF.

Comment:

While some participants were anxious to breastfeed the babies (how anxiety was assessed ?)

Response:

The word ‘anxious’ is rather replace with ‘eager’.

Comment:

"refusal of infants to be breastfed and mother's milk is bad accept" is assessed by vomiting after feeding ???

Response:

The vomit-related phrase has been removed from the sentence.

Comment:

Introduction:

From the introduction we learn very little about the specifics of exclusive breastfeeding in South Africa

Response:

The introduction has been revised to aver the readers about issues of exclusive breastfeeding in South Africa

Comment:

According to Kavle et al. [24], antenatal and postnatal care providers are required to have the most up-to-date and essential skills in order to resolve issues that may arise; hence, adequate training is necessary. In addition, the government must urgently develop and implement policies that protect, encourage, and encourage the EBF as a component of an intervention strategy [14]. - the introduction should explain in more detail the social, cultural, economic, traditional, health and other specificities of breastfeeding in southern Africa, and the quoted sentences should be left for discussion.

Response:

The introduction has been revised as per your suggestion.  In addition, the quoted sentence have been moved to the discussion section accordingly.

Comment:

In light of this, this study investigates the barriers and support systems necessary to support and improve the practice of EBF in this region. - can it be said that the data analysis of the 16 respondents is research into the breastfeeding support system.

Response:

This comment is not clear.

Study setting:

Comment:

From the population of from 47,625 people were selected 16 respondents - is this a representative sample

Response:

This was a qualitative study; therefore, the purpose of qualitative studies is not to generalise per se, but to gain insight into a phenomenon under study. We acknowledged this as a limitation of the study. See the ‘Study limitations’. The purpose of the study's context was to provide readers with an understanding of the unique characteristics of the location where the research was conducted.

Comments:

-According to which criteria were chosen 4 public healthcare facilities to participate in the study.

- from which clinics mothers were chosen to participate in the study - if they were separated from clinics where mothers seek help, then the authors influenced the sample

Response:

We have stated in the ‘Participants and sampling” section that the four public healthcare facilities were conveniently selected.

“The participants were 18-to-40-year-old mothers with infants aged 6 to 12 months who accessed infant and child health care services at four conveniently selected public healthcare facilities in Tswelopele Municipality. Conversely, a purposive sampling was applied to select these mothers because they were likely to have recently completed the period in which EBF may have been practiced, making it simple for them to recall their specific breastfeeding practices”.

Data collection:

Comment:

According to which criteria the questions were formed

Response:

In qualitative research, it is not necessary to cite references for interview questions; however, the interview guide questions were focused on the research objectives.

Results:

Comments:

Table 2. Demographic profile of participants

  1. whether the demographic profile of participants differs from the demographic profile of mothers who did not participate in the research

Response:

This comment is a not clear. However, we were only interested in the demographic characteristics of the mothers who participated in the study.

  1. whether the factors highlighted in Table 2 influenced the responses of the respondents, i.e. whether differences in the responses of the respondents can be determined based on the differences in these factors.

Response:

This comment is not clear.

Comment:

Table 3. Identified themes and sub-themes. - what were the criteria for selecting subtopics - for example, similar answers from more than 50% of respondents.

Theme 1: Mother-related barriers to EBF - whether "having a low breast

Response:

We explained the process of statistical analysis under the data analysis section, which result to the identification of the themes and sub-themes:

 “The process of data analysis was guided by Tesch's eight-step approach [26]. The analysis of the data began with the transcription of the interviews. The initial data set was transcribed by repeatedly perusing it and listening to the audio recordings of the interviews. Then, common themes were identified, and the newly collected data were compared to the previously collected data to determine which themes were confirmed or not backed by the emergent information. The identified themes as well as sub-themes were then categorized, coded, and forwarded to an independent coder for review”.

Comment:

Milk volume" can be explained by "crying and restless infants" or is it a link presented by the respondents? (express more clearly)

Response:

This has been rephrased thus:

“However, once they returned home, they found it difficult to continue EBF for six months for a wide range of factors, including their opinion of having a low breast milk volume, baby cry and restlessness”.

Comment:

Sub-theme 1.1: Extreme breast pain when breastfeeding - were the respondents familiar with the influence of proper breastfeeding technique on the pain during breastfeeding? did they use any procedures after breastfeeding to relieve nipple soreness? 

Response:

Judging from their responses, it appears the mothers were not familiar with the influence of proper breastfeeding technique on the pain during breastfeeding. Whether they used any procedures after breastfeeding to relieve nipple soreness was not probed.

Comment:

Sub-theme 1.2: Mother's desire to introduce solids prematurely - do the authors have information about prenatal education of mothers? are there data in other works about the level of knowledge of mothers about breastfeeding ?

Response:

We have added this in the discussion pertaining to this aspect: “. Studies have shown that mothers who are knowledgeable about the benefits of EBF have a positive attitude toward sustaining exclusive breastfeeding using various methods, such as breast pumps and expressed breast milk [49,50].  As part of South Africa's efforts to encourage and promote exclusive breastfeeding, nurses are required to educate first-time mothers about lactation during antenatal care and labor [24]. Despite the health worker's recommendation to exclusively provide breast milk for six months, mothers feel compelled to introduce solids or other fluids [51]”.

Comment:

Sub-theme 1.3: Mother’s diseases  - pregnancy cannot be equated with illness; they told me they going to put an injection on me - who said he would give an injection that could harm the child.

Response:

The sentence has been corrected to read: “Another significant maternal concern that hindered exclusive breastfeeding was mother’s sickness”.

Comment:

Sub-theme 1.4: Ignorance of the practice of EBF - can the authors describe the differences in the responses of respondents who had breastfeeding education and those who did not have such education.

Response:

The responses of those who have knowledge of the breastfeeding were:

“I think had I known in time that I should give breast milk only and no other drinks and no other foods for first six months of life, I would have avoided to even thinking of trying to give formula milk even to start giving water.” (Participant 7)

“I heard at the hospital when I was going to deliver, they said I must give breast milk only for six months and to never give other foods or drinks including water and formula milk before then.” (Participant 9)

Comment:

Sub-theme 1.5: Traditional beliefs like 'mohlala' - the concept of „mohlala“ and what the health service is doing about it should be explained in more detail

Response:

Mothers are persuaded to believe that breast milk is connected to cultural beliefs or practices, which is known as "mohlala." When an infant is crying excessively while being breastfed, it is a sign of ill omen, and the mother should immediately discontinue breastfeeding.

Comment:

Sub-theme 2.1: Baby's refusal to breastfeed - the authors should definitely describe the "baby's refusal to breastfeed" syndrome in great detail and support it with data from professional literature.

Response:

The baby’s refusal to breastfeed’ has been supported by the literature the in discussion section thus:

“Notably, the potential causes of insufficient breast milk cited by mothers in this study may be attributable to poverty and improper nutrition. Nevertheless, according to the World Health Organization, breast milk could be enhanced by feeding the infant every two to three hours [47], which may be sufficient for the mother and the infant. Similarly, the barriers cited by the mothers in this study, and their belief that the crying of a newborn in the early stages of birth indicate hunger, may mislead the mother into believing that she has insufficient breast milk, resulting in the tendency to introduce other foods or infant formula. It is therefore essential to encourage mothers to develop confidence in their capacity to generate enough milk for their infants [48]. Thus, antenatal settings are ideally suited for such breastfeeding-related talks and discussions to help mothers and healthcare professionals avoid misinterpreting early infant behaviors (crying, short night time sleep, refusal to breastfeed) as pathological during breastfeeding”.

Comment:

Intolerance to breast milk, as evidenced by vomiting after breastfeeding. - authors should very carefully explain "intolerance to breast milk" which is recognized "by vomiting after breastfeeding", and they should support this with data from the literature.

Response:

This is also supported with the literature as indicated above in the discussion. In addition, we understand the concept of literature control in interpretation of the findings. However, we deemed it unnecessary to provide a literature control to support the topics and sub-themes in this study- the implementation of the literature control in this case is somewhat addressed in the discussion section, hence the focus of the discussion is on the results derived from the themes.

Comment:

Theme 3: Support system to enhance EBF -  In the introduction, much more should have been said about the peculiarities of the health system and the way mothers are prepared for pregnancy and breastfeeding, as well as breastfeeding support programs organized in South Africa. In addition, the responses of mothers who have received breastfeeding education should be separated from those who have not. In this way, it is very difficult to evaluate the answers of the respondents.

Response:

We have added in the introduction to highlight the peculiarities of the health system and the way mothers are prepared for pregnancy and breastfeeding, as well as breastfeeding support programs organized in South Africa. “Notably, South Africa had many policies, initiatives, strategies, and recommendations to promote, protect, and support breastfeeding [23]. These include the National Department of Health (NDoH) strong commitment to advance both the Infant and Young Child Feeding Policy (IYCF) and maternal nutrition cause in the country, which led to the 2011 Tshwane Declaration of Support for Breastfeeding in South Africa. The government also revised the IYCF policy to motivate HIV-positive mothers to breastfeed their infants exclusively for the first six months while receiving ART to prevent Mother-To-Child-Transmission (MTCT) and to continue breastfeeding until their babies are one year old [24]. The NDoH Roadmap for Nutrition in South Africa 2013–2017 [25] and the National Strategic Plan for Maternal, Newborn, Child, and Adolescent Health [26] are other initiatives. In addition, nurses were urged to educate first-time mothers about breastfeeding throughout antenatal care and during labour [24]. Other breastfeeding initiatives include Kangaro Mother Care (KMC), MomConnect for pregnant women and mothers [27], the Mother Baby-Friendly Hospital Initiative (MBFHI) [28] launched in 1991, and the KwaZulu-Natal initiative for Breastfeeding Support (KIBS) [29]. Notwithstanding these laudable drives, observably, they fail to account for the influence of family, community, and the workplace [30], and the 2016 South Africa Demographic and Health Survey reports that only 32% of infants younger than six months receive EBF. Moreover, South Africa reportedly has the lowest breastfeeding rate in Africa [31]. Even before the increase in HIV prevalence, EBF was rare in South Africa [32]. Sociocultural, economic, and healthcare system-related concerns are associated with insufficient lactation practices in South Africa. Social conditions, such as lack of income, reliance on food purchases, young mothers' emotions about breastfeeding, and cultural beliefs, were the primary determinants of mothers' breastfeeding behaviour [32-34]. The breastfeeding process excluded fathers. Other studies have alluded to the lack of support and involvement of the father figure in the breastfeeding agenda as an impediment to the lack of EBF and other poor breastfeeding markers in the South Africa [32]. Additionally, conflicting breastfeeding messages from family members/friends, healthcare workers, low socioeconomic indicators, and the fear of transmitting HIV to the newborn infant hinder efforts to exclusively breastfeed in South Africa [35-37]. In this regard, it is crucial that healthcare professionals provide mothers with pertinent EBF education and counselling (13, 9, 38, 39]”.

Comment:

Sub-theme 4.1: Baby’s failure to bond with its mother - rather one could talk about the mother's failure to achieve secure attachment rather than the child's failure. Please support any such claim, regardless of whether it reflects the author's position or the respondents' position, with references

Response:

We have rephrased the theme as “Mother’s failure to bond with the baby”, and added: “When breastfeeding is effortless and uninterrupted, the well-being of both the mother and the infant is typically high [42]. However, premature introduction of solid foods may reduce the amount of time a newborn spends nursing, thereby diminishing the benefits of the unique bond between mother and child”.

Comment:

Some mothers experience breastfeeding challenging and humiliating - breastfeeding is a challenge for every parent, so not only the mother but also the father. We expect the father to support the mother, wanting the best for his child. The authors should explain in what way the test subjects experienced breastfeeding as humiliating.

Response:

The word humiliating has been deleted from the sentence.

Comment:

Few participants reported that nursing was challenging because of painful breasts and nipples - again, the authors do not say anything about the procedures for relieving pain in the nipples or how familiar the test subjects are with these procedure.

Response:

We have added this to the discussion: “Previous research has linked nipple difficulties to poor latching and placement [54,55]; however, a recent study [56] reported a contradictory finding. Instead of seeking alternatives to breastfeeding, mothers should be educated on how to overcome these physical obstacles to EBF by doing the correct thing”.

Comments:

-Authors should distinguish maternal exhaustion caused by breastfeeding from exhaustion due to a disturbed mother's sleep rhythm, exhaustion due to child care (except breastfeeding), exhaustion due to household duties, exhaustion due to care for other children

-If they are already talking about the exhaustion of the mother due to breastfeeding, the authors should also talk about the system of family support for mothers (help of partner, mother, sister, etc.). What are the customs and family roles in the country where the study was conducted?

Responses:

This is added to the discussion: “Child care, domestic chores, and the care of other children may have contributed to the mother's boredom; however, these issues were not investigated. Nevertheless, anecdotal evidence suggests that mothers are rarely assisted with lactation, particularly in terms of assistance with other family responsibilities. At most, they received breastfeeding advice from close family members and relatives, which can be misleading and at incompatible with the principles of good lactation practices”.

Comment:

Mothers can get information about good or poor progress of their child from the pediatrician. If, nevertheless, they associate the infant's crying exclusively with unsuccessful breastfeeding, the authors should consider the option of other possible reasons, which are hidden behind the formal statement that the mother's milk was not sufficient for the child's development. If the system of family, health and social support for breastfeeding is weak, perhaps mothers are looking for an acceptable way to stop breastfeeding, and as a reason for stopping they offer a problem on the part of the child, in order to avoid condemnation from the environment.

Response:

We agreed with your postulation. In this regard, we have revised the discussion thus: “Notably, the potential causes of insufficient breast milk cited by mothers in this study may be attributable to poverty and improper nutrition. Nevertheless, according to the World Health Organization, breast milk could be enhanced by feeding the infant every two to three hours [47], which may be sufficient for the mother and the infant. Similarly, the barriers cited by the mothers in this study, and their belief that the crying of a newborn in the early stages of birth indicate hunger, may mislead the mother into believing that she has insufficient breast milk, resulting in the tendency to introduce other foods or infant formula. It is therefore essential to encourage mothers to develop confidence in their capacity to generate enough milk for their infants [48]. Thus, antenatal settings are ideally suited for such breastfeeding-related talks and discussions to help mothers and healthcare professionals avoid misinterpreting early infant behaviors (crying, short night time sleep, refusal to breastfeed) as pathological during breastfeeding”.

Comment:

It is not enough to say that mothers do not apply the acquired knowledge about exclusive breastfeeding. Possible reasons should be discussed. The reason is probably not the mother's desire to harm the child. Then why doesn't the mother provide the child with the best food. Other research clearly shows that the answer should be sought in the system of family, social, economic and health support. The authors limit themselves only to the immediate, although in conversations with women they were able to find out data for a deeper analysis.

Response:

We have revised the discussion thus: “Studies have shown that mothers who are knowledgeable about the benefits of EBF have a positive attitude toward sustaining exclusive breastfeeding using various methods, such as breast pumps and expressed breast milk [49,50].  As part of South Africa's efforts to encourage and promote exclusive breastfeeding, nurses are required to educate first-time mothers about lactation during antenatal care and labor [24]. Despite the health worker's recommendation to exclusively provide breast milk for six months, mothers feel compelled to introduce solids or other fluids [51]”.

Comment:

Include infants who refuse to breastfeed or who have an intolerance to breast milk - what is the probability that the authors of the 16 respondents encountered children who refuse to breastfeed or who have an intolerance to breast milk. Can the authors say according to the data in the literature that intolerance to breast milk is common?

Response:

We have revised the discussion where we mentioned about misinterpreting early infant behaviors(crying, short nighttime sleep, refusal to breastfeed) as pathological during breastfeeding.

“Notably, the potential causes of insufficient breast milk cited by mothers in this study may be attributable to poverty and improper nutrition. Nevertheless, according to the World Health Organization, breast milk could be enhanced by feeding the infant every two to three hours [47], which may be sufficient for the mother and the infant. Similarly, the barriers cited by the mothers in this study, and their belief that the crying of a newborn in the early stages of birth indicate hunger, may mislead the mother into believing that she has insufficient breast milk, resulting in the tendency to introduce other foods or infant formula. It is therefore essential to encourage mothers to develop confidence in their capacity to generate enough milk for their infants [48]. Thus, antenatal settings are ideally suited for such breastfeeding-related talks and discussions to help mothers and healthcare professionals avoid misinterpreting early infant behaviors (crying, short night time sleep, refusal to breastfeed) as pathological during breastfeeding”.

Comment:

Desai et al. [42] found that the majority of mothers (84%) knew that EBF was essential, particularly during the first six months, because it reduces infants' risk of illness and provides them with sufficient energy for growth. - the authors remain on the surface of the problem, they themselves cite examples of mothers who have been educated about breastfeeding and who stopped breastfeeding anyway. This clearly indicates that mothers' knowledge is not enough for successful breastfeeding. Breastfeeding is a challenge, it is demanding and the mother needs organized and systematic help to persist in breastfeeding, which in most children starts in the maternity ward. It is not enough to just tell mothers how important exclusive breastfeeding is, they need to be supported and helped in an organized way, through the family support system, through visits from the visiting nurse, through breastfeeding support groups, through social and economic support measures organized by state institutions

Response:

We concur with you that mothers' knowledge alone is insufficient for successful breastfeeding, which is also a difficult and burdening endeavour. Consequently, we have highlighted some of the concerns voiced in this context under "Implications of the study." As evidenced by our 'conclusion,' we advocated, from a broader perspective, the implementation of context-specific interventions that are tailored to a geographical context.

Study limitations

We used a purposive sampling technique - it compromises the results

Response:

In this instance, the purposive sampling method is applicable. Moreover, we identified it as a limitation. 

Comment:

Finally, we had originally planned to conduct focus group discussions - this can explain the insufficient depth of the work, the insufficient processing of the respondents' statements, the consistent acceptance of their statements without requiring them to explain these statements, and even when they are contradictory or when the respondents themselves confirm that they acted contrary to the given instructions.

Response:

Being honest is also part of conducting research ethically. Focus group discussions do not necessarily provide more accurate data; they are a component of interviews. In general, one cannot guarantee that interviews of this nature, whether one-on-one or in a focus group, will yield 100 percent accurate information.

Reviewer 2 Report

I congratulate the authors for their qualitative work on supporting the continuation of breastfeeding.

1.      Not questioning the birth order of the children and not knowing gestational duration are limitations.

2.      Related references are absent in some sentences including “In this regard, it is crucial that healthcare professionals provide mothers with pertinent EBF education and counselling.” [ref 36 and “ Oflu A, Yalcin SS, Bukulmez A, Balikoglu P, Celik E. Timely initiation of breastfeeding and its associated factors among Turkish mothers: a mixed model research. Sudan J Paediatr. 2022;22(1):61-69. doi: 10.24911/SJP.106-1616630272.” could be included] 

3.      “Yalçin SS, Berde AS, Yalçin S. Determinants of Exclusive Breast Feeding in sub-Saharan Africa: A Multilevel Approach. Paediatr Perinat Epidemiol. 2016 Sep;30(5):439-49. doi: 10.1111/ppe.12305” reference should be cited in sentence “While EBF is low globally at 35%, the percentage is even lower in Sub-Saharan Africa, ranging from 22% to 33%”

4.      Qualitative research method should be written; in-depth interviews?.

5.      Types of Qualitative research design should be described; Phenomenology, Field research?.

6.      Explain the thematic approach; inductiive or deductive…..

7.      Ref 36 should be cited in related sentences; “Barriers associated with EBF are multifactorial and have been extensively discussed in the literature……..lack of awareness about EBF [8, 13, …..],………., social and cultural factors beliefs and practices (16,3,20,21…….]..” and “Participants also reported that maternal illnesses compelled them to discontinue breastfeeding in order to secure the well-being and health of their infants.

8.      It is necessary to write the references that form the basis of the study questions; Interview guide.

9.      Which statistical program was used in reading the texts and creating the themes?

Minor editing of English language required

Author Response

Reviewer 2

Comments and Suggestions for Authors

I congratulate the authors for their qualitative work on supporting the continuation of breastfeeding.

  1. Not questioning the birth order of the children and not knowing gestational duration are limitations.

Response:

This is included the limitation section thus:  Thirdly, the birth order of the children and the gestational duration were not explored.

  1. Related references are absent in some sentences including “In this regard, it is crucial that healthcare professionals provide mothers with pertinent EBF education and counselling.” [ref 36 and “ Oflu A, Yalcin SS, Bukulmez A, Balikoglu P, Celik E. Timely initiation of breastfeeding and its associated factors among Turkish mothers: a mixed model research. Sudan J Paediatr. 2022;22(1):61-69. doi: 10.24911/SJP.106-161663” could be included] 

Response:

We have added the suggested references and re-arrange the citation numbering in the text accordingly: “In this regard, it is crucial that healthcare professionals provide mothers with pertinent EBF education and counselling (13, 9, 38, 39]”.

  1. “Yalçin SS, Berde AS, Yalçin S. Determinants of Exclusive Breast Feeding in sub-Saharan Africa: A Multilevel Approach. Paediatr Perinat Epidemiol. 2016 Sep;30(5):439-49. doi: 10.1111/ppe.12305” reference should be cited in sentence “While EBF is low globally at 35%, the percentage is even lower in Sub-Saharan Africa, ranging from 22% to 33%”

Response:

We have added the suggested reference and re-arrange the citation numbering in the text accordingly: While EBF is low globally at 35%, the percentage is even lower in Sub-Saharan Africa, ranging from 22% to 33% [9]

  1. Qualitative research method should be written; in-depth interviews?.

Response

This is rephrase thus: “The in-depth interviews took place in specified private rooms within each health facility

  1. Types of Qualitative research design should be described; Phenomenology, Field research?.

Response

The type of qualitative has been added to the methodology section thus:

Research design

A qualitative, explorative and descriptive research design was employed to explore the perspectives of mothers concerning barriers to EBF practices, in addition to the support systems to strengthen exclusive breast-feeding in the context of Tswelopele Municipality.

  1. Explain the thematic approach; inductive or deductive…..

Response

The data were coded using a general inductive technique until no new concepts emerged. This is explained under data analysis section.

  1. Ref 36 should be cited in related sentences; “Barriers associated with EBF are multifactorial and have been extensively discussed in the literature……..lack of awareness about EBF [8, 13, …..],………., social and cultural factors beliefs and practices (16,3,20,21…….]..” and “Participants also reported that maternal illnesses compelled them to discontinue breastfeeding in order to secure the well-being and health of their infants.”

Response

We have included the suggested reference in the above-mentioned sentence and re-arrange the citation numbering in the text accordingly:

“Barriers associated with EBF are multifactorial and have been extensively discussed in the literature. They include unemployment [12], caesarean deliveries [8], lack of awareness about EBF [9,13,14], contradictory family advice on EBF [14-16]. Other impediments to exclusive breastfeeding include the mother's desire to return to full-time employment after maternity leave (3,15,17-19], social and cultural factors beliefs and practices (3, 9, 14,16,20,21,23,36]”.

  1. It is necessary to write the references that form the basis of the study questions; Interview guide.

Response

In qualitative research, it is not necessary to cite references for interview questions; however, the interview guide questions were focused on the research objectives.

  1. Which statistical program was used in reading the texts and creating the themes?

Response

We explained the process of statistical analysis under the data analysis section.

“The process of data analysis was guided by Tesch's eight-step approach [26]. The analysis of the data began with the transcription of the interviews. The initial data set was transcribed by repeatedly perusing it and listening to the audio recordings of the interviews. Then, common themes were identified, and the newly collected data were compared to the previously collected data to determine which themes were confirmed or not backed by the emergent information. The identified themes as well as sub-themes were then categorized, coded, and forwarded to an independent coder for review”.

Comments on the Quality of English Language

Minor editing of English language required

Response

The entire manuscript has been edited, also based on the several editorial suggestions and corrections from one of the reviewers.

Reviewer 3 Report

The topic is interesting and above all it makes the photograph to a really worrying reality because denying breastfeeding is taken away an important opportunity in places where infant mortality is high.

If mothers do not breastfeed exclusively for the first six months of an infant's existence, as recommended by the World Health Organization [5], socio-psychological and clinical health complications could occur for both mother and child.

It should be written in another, less peremptory way. This sentence could be anticipated which results in a repetition: “Furthermore, the World Health Organization [22] endorses the promotion and safeguarding of EBF for the first six months of an infant's life; furthermore, it recommends the addition of complementary foods at six months, along with breastfeeding, as this method has proven to be the most effective feeding practice to prevent infant-related illnesses and mortality worldwide, regardless of the HIV status and socioeconomic status of mothers”.

In order not to repeat this list you could write that the same motivations were found in South Africa

“Barriers associated with EBF are multifactorial and have been extensively discussed in the literature. They include unemployment [12], caesarean deliveries ([8], lack of awareness about EBF [8, 13], contradictory family advice on EBF [14-16] Other impediments to exclusive breastfeeding include the mother's desire to return to full-time employment after maternity leave (17,15,3,18,19], social and cultural factors beliefs and practices (16,3,20,21]”.

“In South Africa, the low percentage is attributable to a combination of variables such as the health care system and family environment, maternal-baby factors, and social and cultural factors such as family pressure and returning to school or work [12,21, 19,20,23]”

Data collection

It is understood that participation is voluntary but it is not clear how these women were chosen.

In the description of the steps of the process it is clear reliability and this is positive.

There is no quantitative assessment in the results. For example, how many have indicated a type of obstacle? In what percentage?

It is said that different foods were given prematurely before 6 months. But how prematurely? At two months, at three months ...? It must be specified. It is also not clear what is given as an alternative to breast milk. For example, it says other liquids. What?  Could it be more specific? I think it's important because it impacts a lot on health. For example, we talk about mashed potatoes but when was it introduced?

Could you explain the economic aspect better? In a condition of poverty, do they still buy formula milk or do they give cow's milk or even other foods?

Are we talking about Prenan? But are they premature babies?

It is said that the Cerelac was given at the suggestion of the grandmother. What product is it and at what age was it given? This story has been written twice.

It should be specified in the discussion that vomiting is an almost physiological condition up to 6 months when the muscular component of the pylorus acquires tone. The important thing is that growth is regular. Isolated vomiting is not a sign of intolerance.

Then the use of water is indicated as highly prejudicial. In fact it is preferable not to give them in quantities that can replace the feeding because it does not give calories that are needed for growth at this stage.

The discussion should be made more fluid.

 cThe topic is interesting and above all it makes the photograph to a really worrying reality because denying breastfeeding is taken away an important opportunity in places where infant mortality is high.

If mothers do not breastfeed exclusively for the first six months of an infant's existence, as recommended by the World Health Organization [5], socio-psychological and clinical health complications could occur for both mother and child.

It should be written in another, less peremptory way. This sentence could be anticipated which results in a repetition: “Furthermore, the World Health Organization [22] endorses the promotion and safeguarding of EBF for the first six months of an infant's life; furthermore, it recommends the addition of complementary foods at six months, along with breastfeeding, as this method has proven to be the most effective feeding practice to prevent infant-related illnesses and mortality worldwide, regardless of the HIV status and socioeconomic status of mothers”.

In order not to repeat this list you could write that the same motivations were found in South Africa

“Barriers associated with EBF are multifactorial and have been extensively discussed in the literature. They include unemployment [12], caesarean deliveries ([8], lack of awareness about EBF [8, 13], contradictory family advice on EBF [14-16] Other impediments to exclusive breastfeeding include the mother's desire to return to full-time employment after maternity leave (17,15,3,18,19], social and cultural factors beliefs and practices (16,3,20,21]”.

“In South Africa, the low percentage is attributable to a combination of variables such as the health care system and family environment, maternal-baby factors, and social and cultural factors such as family pressure and returning to school or work [12,21, 19,20,23]”

Data collection

It is understood that participation is voluntary but it is not clear how these women were chosen.

In the description of the steps of the process it is clear reliability and this is positive.

There is no quantitative assessment in the results. For example, how many have indicated a type of obstacle? In what percentage?

It is said that different foods were given prematurely before 6 months. But how prematurely? At two months, at three months ...? It must be specified. It is also not clear what is given as an alternative to breast milk. For example, it says other liquids. What?  Could it be more specific? I think it's important because it impacts a lot on health. For example, we talk about mashed potatoes but when was it introduced?

Could you explain the economic aspect better? In a condition of poverty, do they still buy formula milk or do they give cow's milk or even other foods?

Are we talking about Prenan? But are they premature babies?

It is said that the Cerelac was given at the suggestion of the grandmother. What product is it and at what age was it given? This story has been written twice.

It should be specified in the discussion that vomiting is an almost physiological condition up to 6 months when the muscular component of the pylorus acquires tone. The important thing is that growth is regular. Isolated vomiting is not a sign of intolerance.

Then the use of water is indicated as highly prejudicial. In fact it is preferable not to give them in quantities that can replace the feeding because it does not give calories that are needed for growth at this stage.

The discussion should be made more fluid.

 The topic is interesting and above all it makes the photograph to a really worrying reality because denying breastfeeding is taken away an important opportunity in places where infant mortality is high.

If mothers do not breastfeed exclusively for the first six months of an infant's existence, as recommended by the World Health Organization [5], socio-psychological and clinical health complications could occur for both mother and child.

It should be written in another, less peremptory way. This sentence could be anticipated which results in a repetition: “Furthermore, the World Health Organization [22] endorses the promotion and safeguarding of EBF for the first six months of an infant's life; furthermore, it recommends the addition of complementary foods at six months, along with breastfeeding, as this method has proven to be the most effective feeding practice to prevent infant-related illnesses and mortality worldwide, regardless of the HIV status and socioeconomic status of mothers”.

In order not to repeat this list you could write that the same motivations were found in South Africa

“Barriers associated with EBF are multifactorial and have been extensively discussed in the literature. They include unemployment [12], caesarean deliveries ([8], lack of awareness about EBF [8, 13], contradictory family advice on EBF [14-16] Other impediments to exclusive breastfeeding include the mother's desire to return to full-time employment after maternity leave (17,15,3,18,19], social and cultural factors beliefs and practices (16,3,20,21]”.

“In South Africa, the low percentage is attributable to a combination of variables such as the health care system and family environment, maternal-baby factors, and social and cultural factors such as family pressure and returning to school or work [12,21, 19,20,23]”

Data collection

It is understood that participation is voluntary but it is not clear how these women were chosen.

In the description of the steps of the process it is clear reliability and this is positive.

There is no quantitative assessment in the results. For example, how many have indicated a type of obstacle? In what percentage?

It is said that different foods were given prematurely before 6 months. But how prematurely? At two months, at three months ...? It must be specified. It is also not clear what is given as an alternative to breast milk. For example, it says other liquids. What?  Could it be more specific? I think it's important because it impacts a lot on health. For example, we talk about mashed potatoes but when was it introduced?

Could you explain the economic aspect better? In a condition of poverty, do they still buy formula milk or do they give cow's milk or even other foods?

Are we talking about Prenan? But are they premature babies?

It is said that the Cerelac was given at the suggestion of the grandmother. What product is it and at what age was it given? This story has been written twice.

It should be specified in the discussion that vomiting is an almost physiological condition up to 6 months when the muscular component of the pylorus acquires tone. The important thing is that growth is regular. Isolated vomiting is not a sign of intolerance.

Then the use of water is indicated as highly prejudicial. In fact it is preferable not to give them in quantities that can replace the feeding because it does not give calories that are needed for growth at this stage.

The discussion should be made more fluid.

 cxThe topic is interesting and above all it makes the photograph to a really worrying reality because denying breastfeeding is taken away an important opportunity in places where infant mortality is high.

If mothers do not breastfeed exclusively for the first six months of an infant's existence, as recommended by the World Health Organization [5], socio-psychological and clinical health complications could occur for both mother and child.

It should be written in another, less peremptory way. This sentence could be anticipated which results in a repetition: “Furthermore, the World Health Organization [22] endorses the promotion and safeguarding of EBF for the first six months of an infant's life; furthermore, it recommends the addition of complementary foods at six months, along with breastfeeding, as this method has proven to be the most effective feeding practice to prevent infant-related illnesses and mortality worldwide, regardless of the HIV status and socioeconomic status of mothers”.

In order not to repeat this list you could write that the same motivations were found in South Africa

“Barriers associated with EBF are multifactorial and have been extensively discussed in the literature. They include unemployment [12], caesarean deliveries ([8], lack of awareness about EBF [8, 13], contradictory family advice on EBF [14-16] Other impediments to exclusive breastfeeding include the mother's desire to return to full-time employment after maternity leave (17,15,3,18,19], social and cultural factors beliefs and practices (16,3,20,21]”.

“In South Africa, the low percentage is attributable to a combination of variables such as the health care system and family environment, maternal-baby factors, and social and cultural factors such as family pressure and returning to school or work [12,21, 19,20,23]”

Data collection

It is understood that participation is voluntary but it is not clear how these women were chosen.

In the description of the steps of the process it is clear reliability and this is positive.

There is no quantitative assessment in the results. For example, how many have indicated a type of obstacle? In what percentage?

It is said that different foods were given prematurely before 6 months. But how prematurely? At two months, at three months ...? It must be specified. It is also not clear what is given as an alternative to breast milk. For example, it says other liquids. What?  Could it be more specific? I think it's important because it impacts a lot on health. For example, we talk about mashed potatoes but when was it introduced?

Could you explain the economic aspect better? In a condition of poverty, do they still buy formula milk or do they give cow's milk or even other foods?

Are we talking about Prenan? But are they premature babies?

It is said that the Cerelac was given at the suggestion of the grandmother. What product is it and at what age was it given? This story has been written twice.

It should be specified in the discussion that vomiting is an almost physiological condition up to 6 months when the muscular component of the pylorus acquires tone. The important thing is that growth is regular. Isolated vomiting is not a sign of intolerance.

Then the use of water is indicated as highly prejudicial. In fact it is preferable not to give them in quantities that can replace the feeding because it does not give calories that are needed for growth at this stage.

The discussion should be made more fluid.

Author Response

Reviewer 3

Comments and Suggestions for Authors

The topic is interesting and above all it makes the photograph to a really worrying reality because denying breastfeeding is taken away an important opportunity in places where infant mortality is high.

Comment:

 If mothers do not breastfeed exclusively for the first six months of an infant's existence, as recommended by the World Health Organization [5], socio-psychological and clinical health complications could occur for both mother and child.

It should be written in another, less peremptory way. This sentence could be anticipated which results in a repetition: “Furthermore, the World Health Organization [22] endorses the promotion and safeguarding of EBF for the first six months of an infant's life; furthermore, it recommends the addition of complementary foods at six months, along with breastfeeding, as this method has proven to be the most effective feeding practice to prevent infant-related illnesses and mortality worldwide, regardless of the HIV status and socioeconomic status of mothers”.

 Response:

“If mothers do not breastfeed exclusively for the first six months of an infant's existence, as recommended by the World Health Organization [5], socio-psychological and clinical health complications could occur for both mother and child”.

Response:

This sentence has been deleted as suggested.

Comment:

In order not to repeat this list you could write that the same motivations were found in South Africa

“Barriers associated with EBF are multifactorial and have been extensively discussed in the literature. They include unemployment [12], caesarean deliveries ([8], lack of awareness about EBF [8, 13], contradictory family advice on EBF [14-16] Other impediments to exclusive breastfeeding include the mother's desire to return to full-time employment after maternity leave (17,15,3,18,19], social and cultural factors beliefs and practices (16,3,20,21, 23]”.

“In South Africa, the low percentage is attributable to a combination of variables such as the health care system and family environment, maternal-baby factors, and social and cultural factors such as family pressure and returning to school or work [12,21, 19,20,23]”

Response:

We have deleted the above sentence pertaining the barriers to EBF in South African context as suggested. However, based on a similar comment from one of the reviewer, we has provided a perspectives on the peculiarities of breastfeeding in South Africa.

“Notably, South Africa had many policies, initiatives, strategies, and recommendations to promote, protect, and support breastfeeding [23]. These include the National Department of Health (NDoH) strong commitment to advance both the Infant and Young Child Feeding Policy (IYCF) and maternal nutrition cause in the country, which led to the 2011 Tshwane Declaration of Support for Breastfeeding in South Africa. The government also revised the IYCF policy to motivate HIV-positive mothers to breastfeed their infants exclusively for the first six months while receiving ART to prevent Mother-To-Child-Transmission (MTCT) and to continue breastfeeding until their babies are one year old [24]. The NDoH Roadmap for Nutrition in South Africa 2013–2017 [25] and the National Strategic Plan for Maternal, Newborn, Child, and Adolescent Health [26] are other initiatives. In addition, nurses were urged to educate first-time mothers about breastfeeding throughout antenatal care and during labour [24]. Other breastfeeding initiatives include Kangaro Mother Care (KMC), MomConnect for pregnant women and mothers [27], the Mother Baby-Friendly Hospital Initiative (MBFHI) [28] launched in 1991, and the KwaZulu-Natal initiative for Breastfeeding Support (KIBS) [29]. Notwithstanding these laudable drives, observably, they fail to account for the influence of family, community, and the workplace [30], and the 2016 South Africa Demographic and Health Survey reports that only 32% of infants younger than six months receive EBF. Moreover, South Africa reportedly has the lowest breastfeeding rate in Africa [31]. Even before the increase in HIV prevalence, EBF was rare in South Africa [32]. Sociocultural, economic, and healthcare system-related concerns are associated with insufficient lactation practices in South Africa. Social conditions, such as lack of income, reliance on food purchases, young mothers' emotions about breastfeeding, and cultural beliefs, were the primary determinants of mothers' breastfeeding behaviour [32-34]. The breastfeeding process excluded fathers. Other studies have alluded to the lack of support and involvement of the father figure in the breastfeeding agenda as an impediment to the lack of EBF and other poor breastfeeding markers in the South Africa [32]. Additionally, conflicting breastfeeding messages from family members/friends, healthcare workers, low socioeconomic indicators, and the fear of transmitting HIV to the newborn infant hinder efforts to exclusively breastfeed in South Africa [35-37]. In this regard, it is crucial that healthcare professionals provide mothers with pertinent EBF education and counselling (13, 9, 38, 39].”

Data collection

Comment:

It is understood that participation is voluntary but it is not clear how these women were chosen.

Response:

We stated this under “Participants and sampling” section thus:

“Conversely, a purposive sampling was applied to select these mothers because they were likely to have recently completed the period in which EBF may have been practiced, making it simple for them to recall their specific breastfeeding practices”

Comment:

In the description of the steps of the process it is clear reliability and this is positive.

Response:

No response needed.

Comment

There is no quantitative assessment in the results. For example, how many have indicated a type of obstacle? In what percentage?

Response:

Given that this was a qualitative study, percentages were not provided.

Comment:

It is said that different foods were given prematurely before 6 months. But how prematurely? At two months, at three months ...? It must be specified. It is also not clear what is given as an alternative to breast milk. For example, it says other liquids. What?  Could it be more specific? I think it's important because it impacts a lot on health. For example, we talk about mashed potatoes but when was it introduced?

Response:

The term premature is understood in the context of exclusive breastfeeding; therefore, it refers to the initiation of any solid or liquid food before six months. This unambiguously demonstrates that anything given to the infant other than breastmilk undermines the WHO-recommended practise of EBF. Some respondents indicated the types of solid and liquid food they fed their infants. See examples:

"…my mom was like, no ways this child doesn't get full with the breast as he should, so she went to buy a sachet of Cerelac and prepare it with some water, she prepared it on a small cup, and then we fed the baby two teaspoons of Cerelac, he ate very well as though he was an old baby and fell asleep after that then he was fine." (Participant 2)

In addition, participants stated that timely education would have enabled them to implement EBF.

"I think had I known in time that I should give breast milk only and no other drinks and no other foods for first six months of life, I would have avoided to even thinking of trying to give formula milk even to start giving water." (Participant 7)

Some, however, did not continue EBF for six months, despite receiving sufficient information during their prenatal visits and labor.

"I heard at the hospital when I was going to deliver, they said I must give breast milk only for six months and to never give other foods or drinks including water and formula milk before then." (Participant 9)

"…I figured out my breast did not have enough milk, and he doesn't get full. So I decided by myself to make the mashed potatoes in addition to breast milk, and when I did give him the mashed potatoes, he would calm down and be normal." (Participant 9)

Comment:

Could you explain the economic aspect better? In a condition of poverty, do they still buy formula milk or do they give cow's milk or even other foods?

Response:

This aspect is highlighted under sub-theme 3.3. “Cost-effectiveness: Mothers saved from buying milk formulas and other solids”. We stated that the participants were concerned about the cost of infant nutrition supplies, and concerning their decision to discontinue breastfeeding, they cited the price of alternative foods to breast milk as their main concern. Certain participants expressed the following:

"They taught us that breastfeeding is best in the first six months. You just give breast milk, especially us unemployed mothers, because with me in my first pregnancy, the father left me, and there was no way I could even afford formula milk." (Participant 11)

"...it was not easy for me to wean him because I wondered what else would my baby eat because I can't even afford formula milk." (Participant 12)

Comment:

Are we talking about Prenan? But are they premature babies?

Response:

Due to the mother's illness, the mother was advised by the health facility to discontinue breastfeeding the infant and replace it with Prenan (formula milk).

Comment:

It is said that the Cerelac was given at the suggestion of the grandmother. What product is it and at what age was it given? This story has been written twice.

Response:

In terms of the research's goal, Cerelac's product was insignificant. The study's focus was not on the nutritional content of infant feeding. The key message was that the EBF was not followed and was violated.  Furthermore, we did not inquire about the age, but it was most likely less than six months, hence the emphasis is on EBF within six months. The detected repetition has been removed.

Comment:

It should be specified in the discussion that vomiting is an almost physiological condition up to 6 months when the muscular component of the pylorus acquires tone. The important thing is that growth is regular. Isolated vomiting is not a sign of intolerance.

Response:

This is included in the discussion. “…These problems may include infants who refuse to breastfeed or who have an intolerance to breast milk, and is vomiting after feeding. However, vomiting is an almost physiological condition up to 6 months when the muscular component of the pylorus acquires tone.

Comment:

Then the use of water is indicated as highly prejudicial. In fact it is preferable not to give them in quantities that can replace the feeding because it does not give calories that are needed for growth at this stage.

Response:

Noted.

Comment

The discussion should be made more fluid.

Response

We have revised some aspects of the discussion. The results regarding EBF barriers arising from each theme are presented and discussed. In addition, findings concerning EBF support strategies are provided. Moreover, we acknowledged the limitations of the investigation. Lastly, the study's implications and conclusion are presented.

Reviewer 4 Report

Thank you for requesting  to provide a review of this article about the barriers regarding exclusive breastfeeding for mothers in Tswelopele Municipality, Free State Province in South Africa.

   The main purpose of the analysis was to examine exclusive breastfeeding (EBF) barriers and support systems in order to promote its practice in Tswelopele Municipality.  The main question adressed in the research was if it is possible to investigate the barriers and support systems, necessary to support and improve the practice of EBF in this region.

   The study is a qualitative study, in which 16 mothers were interviewed regarding breastfeeding issues. The topic is original and relevant in the field and brings usefull knowledge regarding the subject. A comprehensive search strategy was used. The review methodology was comprehensive with screening and data extraction. When it comes to the methodology used, no specific improvements should be considered from my point of view.

   The conclusions are consistent with the evidence and the arguments presented, and they adress properly to the main question which conducted the analysis.

   The references are appropriate and well suited for this kind of study. 

    The tables in the article are very clear and easy to be followed and also the women’s statements are extremely sincere and show the concret issues that make EBF very unconvenient for some of the mothers in the study.

  Regarding the structure and accuracy of the phrases, the manuscript has well structured information, with supported evidence and well structured phrases.

   The manuscript is original and well defined. The results provide an advance in current knowledge. The results are being interpreted appropriately and are significant, as well as the conclusions.

  The study is correctly designed and the analysis is being performed at high standards, so the data are robust enough to draw the conclusion. Surely the paper will attract a wide readership. 

   To conclude, the article is written in a proper way and brings useful information regarding the subject. However I have some issues to add in the lines below, especially regarding the English writting:

Abstract:

Line 10: the fear that, not „the notion that”

Line 14: continued, not „continue”

Line 14: their infants’s, not „there infants’”

Line 18: in order to have sufficient, not „so that they have sufficient”

Line 21-23: „in addition...”. It is not clear what the authors wanted to write. Maybe the phrase should have been written as follows: „in addition, health professionals must be empathic and respectful about the mother’s traditions and cultures and also train mothers and their families on the importance of EBF.”

Introduction

Line 2: This has a lot of advantages both for the mothers and babies, not „This is advantageous for mothers as well as babies”

Line 8: Considering this, not „In considering this”

Line 12: Sub-Saharian, not „Sub-Sahara”

Line 21: „.” after „EBF [14-16]

Data collection

Line 3: „,” before „willingly”

Observing trustworthiness

Line 1: was established by creating, not „was established by establishing”

Line 4: from the participants, not „from participants”

Demographic profile of the participants

Line 1: All the participating mothers, not „All participating mothers”

Line 2: All the participants, not „All participants”

Sub-theme 1.1: Extreme breast pain when breastfeeding

Line 2: The participants’s responses, not „The participant responses”

Line 4: was painful, not „it was painful”

Line 11: while breastfeeding, not „breastfeeding” 

Abstract:

Line 10: the fear that, not „the notion that”

Line 14: continued, not „continue”

Line 14: their infants’s, not „there infants’”

Line 18: in order to have sufficient, not „so that they have sufficient”

Line 21-23: „in addition...”. It is not clear what the authors wanted to write. Maybe the phrase should have been written as follows: „in addition, health professionals must be empathic and respectful about the mother’s traditions and cultures and also train mothers and their families on the importance of EBF.”

Introduction

Line 2: This has a lot of advantages both for the mothers and babies, not „This is advantageous for mothers as well as babies”

Line 8: Considering this, not „In considering this”

Line 12: Sub-Saharian, not „Sub-Sahara”

Line 21: „.” after „EBF [14-16]

Data collection

Line 3: „,” before „willingly”

Observing trustworthiness

Line 1: was established by creating, not „was established by establishing”

Line 4: from the participants, not „from participants”

Demographic profile of the participants

Line 1: All the participating mothers, not „All participating mothers”

Line 2: All the participants, not „All participants”

Sub-theme 1.1: Extreme breast pain when breastfeeding

Line 2: The participants’s responses, not „The participant responses”

Line 4: was painful, not „it was painful”

Line 11: while breastfeeding, not „breastfeeding” 

Author Response

REVIEWER 4

Comments and Suggestions for Authors

Thank you for requesting  to provide a review of this article about the barriers regarding exclusive breastfeeding for mothers in Tswelopele Municipality, Free State Province in South Africa.

   The main purpose of the analysis was to examine exclusive breastfeeding (EBF) barriers and support systems in order to promote its practice in Tswelopele Municipality.  The main question adressed in the research was if it is possible to investigate the barriers and support systems, necessary to support and improve the practice of EBF in this region.

   The study is a qualitative study, in which 16 mothers were interviewed regarding breastfeeding issues. The topic is original and relevant in the field and brings usefull knowledge regarding the subject. A comprehensive search strategy was used. The review methodology was comprehensive with screening and data extraction. When it comes to the methodology used, no specific improvements should be considered from my point of view.

   The conclusions are consistent with the evidence and the arguments presented, and they adress properly to the main question which conducted the analysis.

   The references are appropriate and well suited for this kind of study. 

    The tables in the article are very clear and easy to be followed and also the women’s statements are extremely sincere and show the concret issues that make EBF very unconvenient for some of the mothers in the study.

  Regarding the structure and accuracy of the phrases, the manuscript has well structured information, with supported evidence and well structured phrases.

   The manuscript is original and well defined. The results provide an advance in current knowledge. The results are being interpreted appropriately and are significant, as well as the conclusions.

  The study is correctly designed and the analysis is being performed at high standards, so the data are robust enough to draw the conclusion. Surely the paper will attract a wide readership. 

   To conclude, the article is written in a proper way and brings useful information regarding the subject. However I have some issues to add in the lines below, especially regarding the English writting:

Abstract:

Line 10: the fear that, not „the notion that”

This is corrected in the abstract: ”These include extreme breast discomfort, maternal illness, the fear that milk supply is insufficient, a lack of understanding about EBF, and the influence of different cultural

factors”.

Line 14: continued, not „continue”

This is corrected in the abstract.

Line 14: their infants’s, not „there infants’”

This is corrected in the abstract: ”In addition, while some participants were anxious to breastfeed the babies and continued EBF for a period of six months, their infant’s health issues prevented them from doing so”.

Line 18: in order to have sufficient, not „so that they have sufficient”

This is corrected in the abstract. ”From the prenatal to the postnatal period, mothers and their families should have access to breastfeeding education and counselling in order to have sufficient time to make informed infant nutrition decisions”.

Line 21-23: „in addition...”. It is not clear what the authors wanted to write. Maybe the phrase should have been written as follows: „in addition, health professionals must be empathic and respectful about the mother’s traditions and cultures and also train mothers and their families on the importance of EBF.”

This suggestion is taken and the sentence revised accordingly:  ”In addition, health professionals must be empathic and respectful about the mother’s traditions and cultures and also train mothers and their families on the importance of EBF”.

Introduction

Line 2: This has a lot of advantages both for the mothers and babies, not „This is advantageous for mothers as well as babies”

This is corrected in the introduction as suggested: ”This has a lot advantageous both for the mothers babies”.

Line 8: Considering this, not „In considering this”

This is corrected in as suggested: ”Considering this, initiatives to strengthen the practice of EBF are essential for promoting maternal health outcomes that are favourable”.

Line 12: Sub-Saharian, not „Sub-Sahara”

This is corrected in as suggested: ”While EBF is low globally at 35%, the percentage is even lower in Sub-Saharian Africa,…”

Line 21: „.” after „EBF [14-16]

This was an oversight. Full stop inserted after EBF [14-16].

Data collection

Line 3: „,” before „willingly”

Comma inserted after willingly a suggested.

Observing trustworthiness

Line 1: was established by creating, not „was established by establishing”

This has been corrected as suggested: ”The credibility of the data was established by creating a relationship of trust with the participants by outlining the research objectives and methodology and by interviewing them for an extended period of time”.

Line 4: from the participants, not „from participants”

This has been corrected as suggested: “Multiple techniques were utilized to collect data, including the use of an audio recorder to capture data directly from the participants during interviews”.

Demographic profile of the participants

Line 1: All the participating mothers, not „All participating mothers”

This has been corrected as suggested: ”All the participating mothers were black, and their ages ranged from 18 to 44 years, with the majority falling between 18 and 29 years”.

Line 2: All the participants, not „All participants”

This has been corrected as suggested: ”All the participants were unemployed and out of school, with most having dropped out before completing their matriculation”.

Sub-theme 1.1: Extreme breast pain when breastfeeding

Line 2: The participants’s responses, not „The participant responses”

This has been corrected as suggested : “The participant’s responses regarding breast issues that prevented EBF were as follows:”

Line 4: was painful, not „it was painful”

This has been corrected as suggested : "My breastfeeding experience was painful”.

Line 11: while breastfeeding, not „breastfeeding” 

This has been corrected as suggested: “Some participating mothers reported having difficulty while breastfeeding due to sore breasts and sore nipples while their infants suckled”.

Round 2

Reviewer 1 Report

I stopped with comments (on comments) at Table 3. You did not respond to some of my comments or did not respond adequately. My goal is not to make it difficult for you to publish your work, but to correct some shortcomings, which could confuse readers. So please correct the comments that I have further explained. Then I am willing to continue with the comments.

-------------------------------------------

Comment:

Regardless of the numerous advantages of exclusive breastfeeding (EBF), few mothers practice it (what it means for the authors „ few mothers“ – 44%)

Response:

The sentence has been revised or rephrased thus: “Despite the numerous advantages of exclusive breastfeeding (EBF), it remains less widely practiced, and it is also associated with context-specific obstacles.

---------------------------------------------------------------------------------------------------------------------

Comment:

This study examines exclusive breastfeeding (EBF) barriers and support systems (how the support system was examine?)

Response:

As per qualitative study, it word ‘examines’ is replace with ‘explores’. As stated, individual, semi-structured interviews were conducted to explore the barriers and support systems to promote EBF practice.

Comment 2.:

I think it is more correct to say that the study explores the experiences and opinions of 16 mothers about the barriers and support systems of exclusive breastfeeding. I think so because no direct research has been done on the obstacles or support systems for exclusive breastfeeding. Mothers' experiences were explored.

-----------------------------------------------------------------------------------------------------------------

Comment:

Four major themes emerged (emerged or were selected by the authors?)

Response:

The four themes 'emerged' from the analysis of the data. This term is commonly used in qualitative research reports.

Comment 2.:

Nothing emerged by itself. I advise you to write something like: "Based on the analysis of the collected data, the authors noticed a grouping of opinions around 4 topics, namely..."

------------------------------------------------------------------------------------------------------------------

Comment:

That maternal factors are the primary contributors to the success or failure of this practice (interrogative construction - are maternal factors independent of the influence of other factors?)

Response:

We stated that the results of our study demonstrate that maternal factors are the main contributing factors to the success or failure of exclusive breastfeeding, at least in the setting under consideration. Nonetheless, this question necessitates a quantitative investigation to determine statistically whether maternal factors are independent of other factors influencing EBF.

Comment 2.:

Numerous works link the success of breastfeeding with family, community, health system support factors, etc. In this context, the statement that maternal factors are the primary factors in the failure of exclusive breastfeeding seems incorrect and unfair to mothers. If you really insist on keeping that sentence formation, please add that the "maternal factor" is strongly influenced by other factors. The proof is that with good support in maternity hospitals, 90% of exclusive breastfeeding is achieved. Weak support (social, economic, family, health, etc.) after the mother's discharge from the maternity hospital results in a sharp decline in exclusive breastfeeding.

----------------------------------------------------------------------------------------------------------------

Comment:

While some participants were anxious to breastfeed the babies (how anxiety was assessed ?)

Response:

The word ‘anxious’ is rather replace with ‘eager’.

Comment 2.:

However, you should write that 'anxious' or 'eager' (as you decide) was evaluated based on the assessment of the respondents' answers. This should be written in the text, because questionnaires are often used to assess mental states. Many of these questionnaires are free to use. That is why it should be emphasized in the paper that the authors did not use any questionnaires for the above assessment.

-----------------------------------------------------------------------------------------------------------------

Comment:

"refusal of infants to be breastfed and mother's milk is bad accept" is assessed by vomiting after feeding ???

Response:

The vomit-related phrase has been removed from the sentence.

Comment 2.:

As far as I can see, the sentence has been retained: "Some of these problems included infants' refusal to be breastfed and breast milk being poorly accepted by some infants". If the authors want to keep the terms "refusal to be breastfed" and "breast milk being poorly accepted", please explain them. Suclking mother's milk is a child's instinctive need since the history of mankind. You cannot say that the child refuses to breastfeed, but that in some children breastfeeding is more difficult to establish. Some mothers are more confused, worried, burdened with their own worries. Care for children is more sensitive, more demanding, more gentle. In these circumstances, breastfeeding can be more difficult to establish. It is also not correct to say that some children do not accept breast milk well. This would mean that only mother's milk causes some difficulties for them, which is extremely rare. More precisely, there are problems in harmonizing mother and child in the establishment of breastfeeding and that in these situations the support of health personnel, family, community, etc. is extremely important.

---------------------------------------------------------------------------------------------------------------------

Comment:

Introduction:

From the introduction we learn very little about the specifics of exclusive breastfeeding in South Africa

Response:

The introduction has been revised to aver the readers about issues of exclusive breastfeeding in South Africa

-------------------------------------------------------------------------------------------------------------------

Comment:

According to Kavle et al. [24], antenatal and postnatal care providers are required to have the most up-to-date and essential skills in order to resolve issues that may arise; hence, adequate training is necessary. In addition, the government must urgently develop and implement policies that protect, encourage, and encourage the EBF as a component of an intervention strategy [14]. - the introduction should explain in more detail the social, cultural, economic, traditional, health and other specificities of breastfeeding in southern Africa, and the quoted sentences should be left for discussion.

Response:

The introduction has been revised as per your suggestion.  In addition, the quoted sentence have been moved to the discussion section accordingly.

----------------------------------------------------------------------------------------------------------------

Comment:

In light of this, this study investigates the barriers and support systems necessary to support and improve the practice of EBF in this region. - can it be said that the data analysis of the 16 respondents is research into the breastfeeding support system.

Response:

This comment is not clear.

Comment 2.:

My opinion is that the study did not directly investigate breastfeeding support systems. It would be more correct to say that the research examined mothers' experiences of the breastfeeding support system. Therefore, based on mothers' experiences, it is indirectly concluded about the failures of the breastfeeding support system. Factors that we would consider very important for the success of the breastfeeding support system were not directly isolated and "measured". The clothes were decided indirectly, based on the mothers' ratings. This is fine, but it should be clearly formulated what the subject of the study is, the mothers' experiences, or the measurement of gaps in the breastfeeding support system.

-------------------------------------------------------------------------------------------------------------------

Study setting:

Comment:

From the population of from 47,625 people were selected 16 respondents - is this a representative sample

Response:

This was a qualitative study; therefore, the purpose of qualitative studies is not to generalise per se, but to gain insight into a phenomenon under study. We acknowledged this as a limitation of the study. See the ‘Study limitations’. The purpose of the study's context was to provide readers with an understanding of the unique characteristics of the location where the research was conducted.

Comment 2.:

I fully agree with the above. But then in the discussion you should be more careful with the statements. For example, referring to Jame et al. [43], state that despite the successful initiation of breastfeeding in hospitals after delivery, some mothers experience breastfeeding as a challenge. I think all mothers experience breastfeeding as a challenge. The fact that a high percentage breastfeed in the maternity hospital and give up at home confirms the scheduling in the support system, parental, social and health. If mothers believe that EBF did not effectively satisfy the hunger of infants, then this is a failure of the health workers in the maternity hospital and the follow-up service. If the mother loses interest in breastfeeding due to painful breasts and nipples, then it is the failure of the health service (visiting nurse, pediatrician, family medicine doctor, maternity hospital staff), and not the mother's sole responsibility. That is not clear enough from your discussion. I suggest that you emphasize the mothers' answers more about how much support they received and from whom, when, where...

---------------------------------------------------------------------------------------------------------------

Comments:

-According to which criteria were chosen 4 public healthcare facilities to participate in the study.

- from which clinics mothers were chosen to participate in the study - if they were separated from clinics where mothers seek help, then the authors influenced the sample

Response:

We have stated in the ‘Participants and sampling” section that the four public healthcare facilities were conveniently selected.

“The participants were 18-to-40-year-old mothers with infants aged 6 to 12 months who accessed infant and child health care services at four conveniently selected public healthcare facilities in Tswelopele Municipality. Conversely, a purposive sampling was applied to select these mothers because they were likely to have recently completed the period in which EBF may have been practiced, making it simple for them to recall their specific breastfeeding practices”.

Comment 2.:

It is still not clear whether you singled out mothers from the group of mothers who visited health institutions because of health problems (their own or their child's). Or they are mothers who came for preventive reasons (weighing the child, counseling on breastfeeding). If you singled out mothers from the group of mothers who sought health care, you directly influenced the sample. Why? Because you singled out only mothers who had problems, the result might have been quite different if you had selected only mothers who had no need for health care. It is important that you write it so that it is clear to every reader, because it is important information. If you took a sample of women who reported to the health system because of health problems, state this in the limitations of the study.

----------------------------------------------------------------------------------------------------------

Data collection:

Comment:

According to which criteria the questions were formed

Response:

In qualitative research, it is not necessary to cite references for interview questions; however, the interview guide questions were focused on the research objectives.

Comment 2.:

Dear colleagues, it is not a matter of whether the study is qualitative or quantitative, but whether all questions are in accordance with the views of the profession. You wrote: "intolerance to mother's milk, which is manifested by vomiting after feeding". I have no experience that vomiting after breastfeeding is an intolerance to breast milk. Intolerance to mother's milk would represent intolerance to the proteins in mother's milk. It is very rare. If there is an intolerance to mother's milk, vomiting is certainly not the only symptom. Conversely, vomiting after breastfeeding is very rarely caused by intolerance to the ingredients of breast milk. So I am asking for a reference for your claim that in your sample of 16 women you had women who stopped breastfeeding because of the child's intolerance to breast milk, which you diagnosed based on the child's vomiting.

---------------------------------------------------------------------------------------------------------------------

Results:

Comments:

Table 2. Demographic profile of participants

a)     whether the demographic profile of participants differs from the demographic profile of mothers who did not participate in the research

Response:

This comment is a not clear. However, we were only interested in the demographic characteristics of the mothers who participated in the study.

b)     whether the factors highlighted in Table 2 influenced the responses of the respondents, i.e. whether differences in the responses of the respondents can be determined based on the differences in these factors.

Response:

This comment is not clear.

Comment 2.:

a)     You should comment on whether the sociodemographic characteristics of your respondents are similar or different from those expected for the environment in which they live. In other words, do the characteristics of the respondents in your sample differ significantly from the characteristics of the population. For example, you have many people tagged as "Single". Is this common in your country? Most mothers are unmarried at the time of their child's birth? Do they live in unmarried homes or are they without the support of a partner? Is this the case only with the respondents from your sample? Educational status is low. Is this a characteristic of all young women in your country? Is it the same with young men? Are your respondents different from young women in the rest of the population? Why?

b)     I am a little confused by the misunderstanding of my comments. I am asking if there are differences in the responses of the respondents according to whether they are married/unmarried and according to whether they have more or less than the legally required basic education (in my country it is eight years)

------------------------------------------------------------------------------

------------------------------------------------------------------------------

Comment:

Table 3. Identified themes and sub-themes. - what were the criteria for selecting subtopics - for example, similar answers from more than 50% of respondents.

Theme 1: Mother-related barriers to EBF - whether "having a low breast

Response:

We explained the process of statistical analysis under the data analysis section, which result to the identification of the themes and sub-themes:

 “The process of data analysis was guided by Tesch's eight-step approach [26]. The analysis of the data began with the transcription of the interviews. The initial data set was transcribed by repeatedly perusing it and listening to the audio recordings of the interviews. Then, common themes were identified, and the newly collected data were compared to the previously collected data to determine which themes were confirmed or not backed by the emergent information. The identified themes as well as sub-themes were then categorized, coded, and forwarded to an independent coder for review”.

Comment 2.:

To the question of whether the subtopics are defined by a certain percentage of similar (common) answers (eg more than 50% of respondents), you answer by quoting what was already written in the text: "The data analysis process was guided by Tesch's eight-step approach [26]. Data analysis began with interview transcription. The initial data set was transcribed by repeatedly reviewing and listening to the audio recordings of the interviews. Common themes were then identified, and newly collected data were compared to previously collected data to determine which themes were confirmed or not supported by the new information. Identified themes as well as sub-themes were then categorized, coded and forwarded to an independent coder for review”. In quoting the sentence that "the subtopics were categorized, coded and forwarded to an independent coder for review" the answer to my question was not given. I did not ask who selected the subtopics, but what was the criteria for selecting subtopics. Specifically, what level of commonality is the criterion for subtopic formation.

Author Response

Reviewer 1

Comments and Suggestions for Authors

I stopped with comments (on comments) at Table 3. You did not respond to some of my comments or did not respond adequately. My goal is not to make it difficult for you to publish your work, but to correct some shortcomings, which could confuse readers. So please correct the comments that I have further explained. Then I am willing to continue with the comments.

 -------------------------------------------

Comment:

Regardless of the numerous advantages of exclusive breastfeeding (EBF), few mothers practice it (what it means for the authors „ few mothers“ – 44%)

Response:

The sentence has been revised or rephrased thus: “Despite the numerous advantages of exclusive breastfeeding (EBF), it remains less widely practiced, and it is also associated with context-specific obstacles.”

---------------------------------------------------------------------------------------------------------------------

Comment:

This study examines exclusive breastfeeding (EBF) barriers and support systems (how the support system was examine?)

Response:

As per qualitative study, it word ‘examines’ is replace with ‘explores’. As stated, individual, semi-structured interviews were conducted to explore the barriers and support systems to promote EBF practice.

Comment 2.:

I think it is more correct to say that the study explores the experiences and opinions of 16 mothers about the barriers and support systems of exclusive breastfeeding. I think so because no direct research has been done on the obstacles or support systems for exclusive breastfeeding. Mothers' experiences were explored.

Response 2:

The suggestion has been taken and the sentence revised accordingly.

“The study explores the experiences and opinions of 16 mothers about the barriers and support systems of exclusive breastfeeding”

-----------------------------------------------------------------------------------------------------------------

Comment:

Four major themes emerged (emerged or were selected by the authors?)

Response:

The four themes 'emerged' from the analysis of the data. This term is commonly used in qualitative research reports.

Comment 2.:

Nothing emerged by itself. I advise you to write something like: "Based on the analysis of the collected data, the authors noticed a grouping of opinions around 4 topics, namely..."

Response 2:

Your suggestion taken: “The analysis of the collected data revealed that opinions clustered around four topics: mother-related barriers to EBF, baby-related barriers to EBF, support systems to enhance EBF, and complications caused by barriers to EBF”.

------------------------------------------------------------------------------------------------------------------

Comment:

That maternal factors are the primary contributors to the success or failure of this practice (interrogative construction - are maternal factors independent of the influence of other factors?)

Response:

We stated that the results of our study demonstrate that maternal factors are the main contributing factors to the success or failure of exclusive breastfeeding, at least in the setting under consideration. Nonetheless, this question necessitates a quantitative investigation to determine statistically whether maternal factors are independent of other factors influencing EBF.

Comment 2.:

Numerous works link the success of breastfeeding with family, community, health system support factors, etc. In this context, the statement that maternal factors are the primary factors in the failure of exclusive breastfeeding seems incorrect and unfair to mothers. If you really insist on keeping that sentence formation, please add that the "maternal factor" is strongly influenced by other factors. The proof is that with good support in maternity hospitals, 90% of exclusive breastfeeding is achieved. Weak support (social, economic, family, health, etc.) after the mother's discharge from the maternity hospital results in a sharp decline in exclusive breastfeeding.

Response 2:

The sentence rephrased as suggested.

The findings from these themes and sub-themes imply that maternal factor is strongly influenced by other factors to the success or failure of this practice.

----------------------------------------------------------------------------------------------------------------

Comment:

While some participants were anxious to breastfeed the babies (how anxiety was assessed ?)

Response:

The word ‘anxious’ is rather replace with ‘eager’.

Comment 2.:

However, you should write that 'anxious' or 'eager' (as you decide) was evaluated based on the assessment of the respondents' answers. This should be written in the text, because questionnaires are often used to assess mental states. Many of these questionnaires are free to use. That is why it should be emphasized in the paper that the authors did not use any questionnaires for the above assessment.

Response 2:

We have stated this in the limitation: “To ascertain this, additional research and alternative (quantitative) methods are necessary”. In addition, the research design is suggestive evidence that this is a qualitative study that does not require a questionnaire.

-----------------------------------------------------------------------------------------------------------------

Comment:

"refusal of infants to be breastfed and mother's milk is bad accept" is assessed by vomiting after feeding ???

Response:

The vomit-related phrase has been removed from the sentence.

Comment 2.:

As far as I can see, the sentence has been retained: "Some of these problems included infants' refusal to be breastfed and breast milk being poorly accepted by some infants". If the authors want to keep the terms "refusal to be breastfed" and "breast milk being poorly accepted", please explain them. Suclking mother's milk is a child's instinctive need since the history of mankind. You cannot say that the child refuses to breastfeed, but that in some children breastfeeding is more difficult to establish. Some mothers are more confused, worried, burdened with their own worries. Care for children is more sensitive, more demanding, more gentle. In these circumstances, breastfeeding can be more difficult to establish. It is also not correct to say that some children do not accept breast milk well. This would mean that only mother's milk causes some difficulties for them, which is extremely rare. More precisely, there are problems in harmonizing mother and child in the establishment of breastfeeding and that in these situations the support of health personnel, family, community, etc. is extremely important.

Response 2:

The sentence has been revised thus: “…their infant’s health and behavioral issues prevented them from doing so. Some of these problems included infants’ sickness and crying”.

---------------------------------------------------------------------------------------------------------------------

Comment:

Introduction:

From the introduction we learn very little about the specifics of exclusive breastfeeding in South Africa

Response:

The introduction has been revised to aver the readers about issues of exclusive breastfeeding in South Africa

-------------------------------------------------------------------------------------------------------------------

Comment:

According to Kavle et al. [24], antenatal and postnatal care providers are required to have the most up-to-date and essential skills in order to resolve issues that may arise; hence, adequate training is necessary. In addition, the government must urgently develop and implement policies that protect, encourage, and encourage the EBF as a component of an intervention strategy [14]. - the introduction should explain in more detail the social, cultural, economic, traditional, health and other specificities of breastfeeding in southern Africa, and the quoted sentences should be left for discussion.

Response:

The introduction has been revised as per your suggestion.  In addition, the quoted sentence have been moved to the discussion section accordingly.

----------------------------------------------------------------------------------------------------------------

Comment:

In light of this, this study investigates the barriers and support systems necessary to support and improve the practice of EBF in this region. - can it be said that the data analysis of the 16 respondents is research into the breastfeeding support system.

Response:

This comment is not clear.

Comment 2.:

My opinion is that the study did not directly investigate breastfeeding support systems. It would be more correct to say that the research examined mothers' experiences of the breastfeeding support system. Therefore, based on mothers' experiences, it is indirectly concluded about the failures of the breastfeeding support system. Factors that we would consider very important for the success of the breastfeeding support system were not directly isolated and "measured". The clothes were decided indirectly, based on the mothers' ratings. This is fine, but it should be clearly formulated what the subject of the study is, the mothers' experiences, or the measurement of gaps in the breastfeeding support system.

Response 2:

Noted. The suggestion has been taken and the sentence revised accordingly.

“In light of this, this study explores the experiences and opinions of mothers about the barriers and support systems of exclusive breastfeeding to improve the practice of EBF in this region”.

-------------------------------------------------------------------------------------------------------------------

Study setting:

Comment:

From the population of from 47,625 people were selected 16 respondents - is this a representative sample

Response:

This was a qualitative study; therefore, the purpose of qualitative studies is not to generalise per se, but to gain insight into a phenomenon under study. We acknowledged this as a limitation of the study. See the ‘Study limitations’. The purpose of the study's context was to provide readers with an understanding of the unique characteristics of the location where the research was conducted.

Comment 2.:

I fully agree with the above. But then in the discussion you should be more careful with the statements. For example, referring to Jame et al. [43], state that despite the successful initiation of breastfeeding in hospitals after delivery, some mothers experience breastfeeding as a challenge. I think all mothers experience breastfeeding as a challenge. The fact that a high percentage breastfeed in the maternity hospital and give up at home confirms the scheduling in the support system, parental, social and health. If mothers believe that EBF did not effectively satisfy the hunger of infants, then this is a failure of the health workers in the maternity hospital and the follow-up service. If the mother loses interest in breastfeeding due to painful breasts and nipples, then it is the failure of the health service (visiting nurse, pediatrician, family medicine doctor, maternity hospital staff), and not the mother's sole responsibility. That is not clear enough from your discussion. I suggest that you emphasize the mothers' answers more about how much support they received and from whom, when, where...

Response 2:

Your comments or observations are appropriate. However, breastfeeding is a behavioural construct that appears to be influenced by a number of opposing factors. Assuming that health workers provide the desired knowledge on breastfeeding practises at health facilities, mothers may be influenced by their preconceived cultural or traditional beliefs or family/close relatives' influence or understanding of the breastfeeding practises. These issues are readily apparent in our research and are emphasised in the results and discussion. Consequently, based on your suggestion to clarify the discussion, we have added these excerpts to the discussion.

This resonates with the findings of this study, some mothers did not maintain breastfeeding for six months despite receiving sufficient information from nurses during prenatal visits and labor at health clinics. However, understanding EBF does not automatically result in mothers practicing it". As earlier indicated by one of the participants: I heard at the hospital when I was going to deliver, they said I must give breast milk only for six months and to never give other foods or drinks including water and formula milk before then." (Participant 9)."…I figured out my breast did not have enough milk, and he doesn't get full. So I decided by myself to make the mashed potatoes in addition to breast milk, and when I did give him the mashed potatoes, he would calm down and be normal." (Participant 9).

---------------------------------------------------------------------------------------------------------------

Comments:

-According to which criteria were chosen 4 public healthcare facilities to participate in the study.

- from which clinics mothers were chosen to participate in the study - if they were separated from clinics where mothers seek help, then the authors influenced the sample

Response:

We have stated in the ‘Participants and sampling” section that the four public healthcare facilities were conveniently selected.

“The participants were 18-to-40-year-old mothers with infants aged 6 to 12 months who accessed infant and child health care services at four conveniently selected public healthcare facilities in Tswelopele Municipality. Conversely, a purposive sampling was applied to select these mothers because they were likely to have recently completed the period in which EBF may have been practiced, making it simple for them to recall their specific breastfeeding practices”.

Comment 2.:

It is still not clear whether you singled out mothers from the group of mothers who visited health institutions because of health problems (their own or their child's). Or they are mothers who came for preventive reasons (weighing the child, counseling on breastfeeding). If you singled out mothers from the group of mothers who sought health care, you directly influenced the sample. Why? Because you singled out only mothers who had problems, the result might have been quite different if you had selected only mothers who had no need for health care. It is important that you write it so that it is clear to every reader, because it is important information. If you took a sample of women who reported to the health system because of health problems, state this in the limitations of the study.

Response: 2

As we earlier stated the participants were mothers accessing infant and child health care services. They were mothers who had visited health facilities for routine infant healthcare screening and maternal counselling in the selected study sites.

----------------------------------------------------------------------------------------------------------

Data collection:

Comment:

According to which criteria the questions were formed

Response:

In qualitative research, it is not necessary to cite references for interview questions; however, the interview guide questions were focused on the research objectives.

Comment 2.:

Dear colleagues, it is not a matter of whether the study is qualitative or quantitative, but whether all questions are in accordance with the views of the profession. You wrote: "intolerance to mother's milk, which is manifested by vomiting after feeding". I have no experience that vomiting after breastfeeding is an intolerance to breast milk. Intolerance to mother's milk would represent intolerance to the proteins in mother's milk. It is very rare. If there is an intolerance to mother's milk, vomiting is certainly not the only symptom. Conversely, vomiting after breastfeeding is very rarely caused by intolerance to the ingredients of breast milk. So I am asking for a reference for your claim that in your sample of 16 women you had women who stopped breastfeeding because of the child's intolerance to breast milk, which you diagnosed based on the child's vomiting.

Response 2:

This usage of "intolerance to mother's milk" refers to infants who lose interest in breastfeeding, which may be caused by a number of factors, including infant illness, crying, restlessness, and mother’s health condition. This has been corrected in the paper. Likewise, the vomiting related issue too has been corrected accordingly.

---------------------------------------------------------------------------------------------------------------------

Results:

Comments:

Table 2. Demographic profile of participants

  1. a)     whether the demographic profile of participants differs from the demographic profile of mothers who did not participate in the research

Response:

This comment is a not clear. However, we were only interested in the demographic characteristics of the mothers who participated in the study.

  1. b)     whether the factors highlighted in Table 2 influenced the responses of the respondents, i.e. whether differences in the responses of the respondents can be determined based on the differences in these factors.

Response:

This comment is not clear.

Comment 2.:

  1. You should comment on whether the sociodemographic characteristics of your respondents are similar or different from those expected for the environment in which they live. In other words, do the characteristics of the respondents in your sample differ significantly from the characteristics of the population. For example, you have many people tagged as "Single". Is this common in your country? Most mothers are unmarried at the time of their child's birth? Do they live in unmarried homes or are they without the support of a partner? Is this the case only with the respondents from your sample? Educational status is low. Is this a characteristic of all young women in your country? Is it the same with young men? Are your respondents different from young women in the rest of the population? Why?

Response 2:

As indicated in the "Study setting" section, the socio-demographic characteristics of the respondents in our study are comparable to those of the population. “Notably, the socio-demographic characteristics of the respondents in our study are comparable to those of the population. The majority of rural women in the study context are poor and had a low level of educational attainment. In addition, their primary occupation is subsistence farming, and the majority of them rely on a government grant or stipend of R500 (approximately USD 29) per month as their household income. Probably because the majority of them are young and likely dropped out of school due to pregnancy, they are either unmarried or residing with their partners. According to Hall et al. [42], South Africa has a significant percentage of single mothers, with just over 60% of children born in 2017 having no registered father”. The above information has been added to the demographic profile of the participants.

  1. I am a little confused by the misunderstanding of my comments. I am asking if there are differences in the responses of the respondents according to whether they are married/unmarried and according to whether they have more or less than the legally required basic education (in my country it is eight years)

Response 2:

Their responses could be similar. We did not, however, compare their responses based on their marital status, level of education, or any other variable. In South Africa, Grade 12 is the minimum requirement for completing secondary education. As evidenced by our sample, the vast majority of them could not complete secondary education. We have added this observation to the 2nd paragraph of the discussion as an opening statement to the discussion of the findings.

“Observably, the low level of education, impoverished economic status, and marital status (majority were single) of the mothers in the present study may have influenced their breastfeeding practices”.

------------------------------------------------------------------------------

Comment:

Table 3. Identified themes and sub-themes. - what were the criteria for selecting subtopics - for example, similar answers from more than 50% of respondents.

Theme 1: Mother-related barriers to EBF - whether "having a low breast

Response:

We explained the process of statistical analysis under the data analysis section, which result to the identification of the themes and sub-themes:

 “The process of data analysis was guided by Tesch's eight-step approach [26]. The analysis of the data began with the transcription of the interviews. The initial data set was transcribed by repeatedly perusing it and listening to the audio recordings of the interviews. Then, common themes were identified, and the newly collected data were compared to the previously collected data to determine which themes were confirmed or not backed by the emergent information. The identified themes as well as sub-themes were then categorized, coded, and forwarded to an independent coder for review”.

 Comment 2.:

 To the question of whether the subtopics are defined by a certain percentage of similar (common) answers (eg more than 50% of respondents), you answer by quoting what was already written in the text: "The data analysis process was guided by Tesch's eight-step approach [26]. Data analysis began with interview transcription. The initial data set was transcribed by repeatedly reviewing and listening to the audio recordings of the interviews. Common themes were then identified, and newly collected data were compared to previously collected data to determine which themes were confirmed or not supported by the new information. Identified themes as well as sub-themes were then categorized, coded and forwarded to an independent coder for review”. In quoting the sentence that "the subtopics were categorized, coded and forwarded to an independent coder for review" the answer to my question was not given. I did not ask who selected the subtopics, but what was the criteria for selecting subtopics. Specifically, what level of commonality is the criterion for subtopic formation.

Response 2:

The criterion for subtopic formation was based on the study research objective and context (EBF barriers). After coding the data, codes with comparable or shared characteristics were categorised.

In other words, based on the topics from the interview guides, a coding framework was designed deductively. Then, codes were re-coded, based on the major themes of the study, to form categories.

Reviewer 2 Report

Thank you authors for their effort. However, comments should be supported by existing references. Please give suitable references for your sentences including; 

"As previously mentioned, research indicates that maternal factors such as drowsiness and fatigue can impede exclusive breastfeeding in a disproportionately high number of cases."

"antenatal settings are ideally suited for such breastfeeding-related talks and discussions to help mothers and healthcare professionals avoid misinterpreting early infant behaviors (crying, short night time sleep, refusal to breastfeed) as pathological during breastfeeding"

"In the majority of African societies where indigenous cultural beliefs are deeply ingrained and respected, women are knitted to men and other family mem-bers and relatives (husbands, mothers, mothers-in-law, grandmothers, and acquaint-ances). In addition, there is typically no formal lactation environment; thus, mothers are free to breastfeed at any time and place. Furthermore, women who breastfeed publicly are not obligated to conceal their breasts. Similarly, a woman's status or respect is determined by her ability to bear and raise children."

"However, vomiting is an almost physiological condition up to 6 months when the muscular component of the pylorus acquires tone."

"Implications of the study" section also needs reference support. 

Minor editing of English language required

Author Response

Reviewer 2

Comments and Suggestions for Authors

Thank you authors for their effort. However, comments should be supported by existing references. Please give suitable references for your sentences including; 

Comment:

"As previously mentioned, research indicates that maternal factors such as drowsiness and fatigue can impede exclusive breastfeeding in a disproportionately high number of cases."

Response

See below and in the text:

“As previously mentioned, research indicates that maternal factors such as drowsiness and fatigue can impede exclusive breastfeeding in a disproportionately high number of cases [45]”.

Comment:

"antenatal settings are ideally suited for such breastfeeding-related talks and discussions to help mothers and healthcare professionals avoid misinterpreting early infant behaviors (crying, short night time sleep, refusal to breastfeed) as pathological during breastfeeding"

Response

See below and in the text:

“Thus, antenatal settings are ideally suited for such breastfeeding-related talks and discussions to help mothers and healthcare professionals avoid misinterpreting early infant behaviors (crying, short night time sleep, reluctance to be breastfeed) as pathological during breastfeeding [50,51]”.

Comment:

"In the majority of African societies where indigenous cultural beliefs are deeply ingrained and respected, women are knitted to men and other family mem-bers and relatives (husbands, mothers, mothers-in-law, grandmothers, and acquaint-ances). In addition, there is typically no formal lactation environment; thus, mothers are free to breastfeed at any time and place. Furthermore, women who breastfeed publicly are not obligated to conceal their breasts. Similarly, a woman's status or respect is determined by her ability to bear and raise children."

Response

See below and in the text:

“The majority of African societies are unfamiliar with EBF because it contradicts their deeply held cultural beliefs and customs [58]. Given that women are knitted with men and other family members and acquaintances, EBF decisions should be made with their input [58]. In addition, there is typically no formal lactation environment; thus, mothers are free to breastfeed at any time and place [59]. Furthermore, in some societies, women who breastfeed publicly are not obligated to conceal their breasts [59]”.

Comment:

"However, vomiting is an almost physiological condition up to 6 months when the muscular component of the pylorus acquires tone."

Response

This sentence has been deleted based on the comment from one of the reviewer.

Comment:

"Implications of the study" section also needs reference support. 

Response

See below and in the text:

“Empowering and supporting mothers through explicit and specific health education regarding the practice, the duration, and the advantages of EBF may increase the uptake of EBF practice in women [70]“.

“Furthermore, family involvement is essential because a supportive companion, relative, or friend is essential for breastfeeding success [71, 72].”

“Therefore, providing consistent and continuing health education on breastfeeding to family members (grandparents, elders, partners, and siblings) and friends will allow other family members who have breastfed to discuss their personal experiences while creating a breastfeeding-friendly environment for nursing mothers [38]”.

Comments on the Quality of English Language

Minor editing of English language required

Response:

The manuscript has been edited where necessary.

Reviewer 4 Report

  Thank you for requesting  to provide a review of this already revised article, about the barriers regarding exclusive breastfeeding for mothers in Tswelopele Municipality, Free State Province in South Africa.

   The main purpose of the analysis was to examine exclusive breastfeeding (EBF) barriers and support systems in order to promote its practice in Tswelopele Municipality.  The main question adressed in the research was if it is possible to investigate the barriers and support systems, necessary to support and improve the practice of EBF in this region.

   The study is a qualitative study. The topic is original and relevant in the field and brings usefull knowledge regarding the subject. A comprehensive search strategy was used. The review methodology was comprehensive with screening and data extraction. When it comes to the methodology used, no specific improvements should be considered from my point of view.

   The conclusions are consistent with the evidence and the arguments presented, and they adress properly to the main question which conducted the analysis.

   The references are appropriate and well suited for this kind of study. 

    The tables in the article are very clear and easy to be followed and also the women’s statements are extremely sincere and show the concret issues that make EBF very unconvenient for some of the mothers in the study.

  Regarding the structure and accuracy of the phrases, the manuscript has well structured information, with supported evidence and well structured phrases.

   The manuscript is original and well defined. The results provide an advance in current knowledge. The results are being interpreted appropriately and are significant, as well as the conclusions.

  The study is correctly designed and the analysis is being performed at high standards, so the data are robust enough to draw the conclusion. Surely the paper will attract a wide readership. 

   To conclude, the article is now written in a proper way and brings useful information regarding the subject. I only have two issues to add in the lines below, but the article is suitable enough to warrant publication in Children.

 INTRODUCTION

Line 1: this has a lot of advantages both for the mother and the baby, not „this has a lot advantageous both for the mother’s babies”

DATA COLLECTION

Line 1: Sixteen participants were selected for individual semi-structured interviews, not „Sixteen participants participated in individual semi-structured interviews”

Line 1: Sixteen participants were selected for individual semi-structured interviews, not „Sixteen participants participated in individual semi-structured interviews”

Author Response

Reviewer 4

Comments and Suggestions for Authors

 Thank you for requesting  to provide a review of this already revised article, about the barriers regarding exclusive breastfeeding for mothers in Tswelopele Municipality, Free State Province in South Africa.

   The main purpose of the analysis was to examine exclusive breastfeeding (EBF) barriers and support systems in order to promote its practice in Tswelopele Municipality.  The main question adressed in the research was if it is possible to investigate the barriers and support systems, necessary to support and improve the practice of EBF in this region.

   The study is a qualitative study. The topic is original and relevant in the field and brings usefull knowledge regarding the subject. A comprehensive search strategy was used. The review methodology was comprehensive with screening and data extraction. When it comes to the methodology used, no specific improvements should be considered from my point of view.

   The conclusions are consistent with the evidence and the arguments presented, and they adress properly to the main question which conducted the analysis.

   The references are appropriate and well suited for this kind of study. 

    The tables in the article are very clear and easy to be followed and also the women’s statements are extremely sincere and show the concret issues that make EBF very unconvenient for some of the mothers in the study.

  Regarding the structure and accuracy of the phrases, the manuscript has well structured information, with supported evidence and well structured phrases.

   The manuscript is original and well defined. The results provide an advance in current knowledge. The results are being interpreted appropriately and are significant, as well as the conclusions.

  The study is correctly designed and the analysis is being performed at high standards, so the data are robust enough to draw the conclusion. Surely the paper will attract a wide readership. 

   To conclude, the article is now written in a proper way and brings useful information regarding the subject. I only have two issues to add in the lines below, but the article is suitable enough to warrant publication in Children.

INTRODUCTION

Comment:

Line 1: this has a lot of advantages both for the mother and the baby, not „this has a lot advantageous both for the mother’s babies”

Response:

The sentence has been revised as suggested.

This has a lot of advantages both for the mother and the baby

DATA COLLECTION

Comment:

Line 1: Sixteen participants were selected for individual semi-structured interviews, not „Sixteen participants participated in individual semi-structured interviews”

Response:

The sentence has been revised as suggested.

Sixteen participants were selected for individual semi-structured interviews.
